# Graph-less Neural Networks: Teaching Old MLPs New Tricks via Distillation

**Shichang Zhang**[*]
University of California, Los Angeles
shichang@cs.ucla.edu

**Yozen Liu**
Snap Inc.
yliu2@snap.com

**Yizhou Sun**
University of California, Los Angeles
yzsun@cs.ucla.edu

**Neil Shah**
Snap Inc.
nshah@snap.com

## ABSTRACT

Graph Neural Networks (GNNs) are popular for graph machine learning and have shown great results on wide node classification tasks. Yet, they are less popular for practical deployments in the industry owing to their scalability challenges incurred by data dependency. Namely, GNN inference depends on neighbor nodes multiple hops away from the target, and fetching them burdens latency-constrained applications. Existing inference acceleration methods like pruning and quantization can speed up GNNs by reducing Multiplication-and-ACcumulation (MAC) operations, but the improvements are limited given the data dependency is not resolved. Conversely, multi-layer perceptrons (MLPs) have no graph dependency and infer much faster than GNNs, even though they are less accurate than GNNs for node classification in general. Motivated by these complementary strengths and weaknesses, we bring GNNs and MLPs together via knowledge distillation (KD). Our work shows that the performance of MLPs can be improved by large margins with GNN KD. We call the distilled MLPs Graph-less Neural Networks (GLNNs) as they have no inference graph dependency. We show that GLNNs with competitive accuracy infer faster than GNNs by 146×-273× and faster than other acceleration methods by 14×-27×. Under a production setting involving both transductive and inductive predictions across 7 datasets, GLNN accuracies improve over stand-alone MLPs by 12.36% on average and match GNNs on 6/7 datasets. Comprehensive analysis shows when and why GLNNs can achieve competitive accuracies to GNNs and suggests GLNN as a handy choice for latency-constrained applications.

## 1 Introduction

Graph Neural Networks (GNNs) have recently become very popular for graph machine learning (GML) research and have shown great results on node classification tasks (Kipf & Welling, 2016; Hamilton et al., 2017; Veličković et al., 2017) like product prediction on co-purchasing graphs and paper category prediction on citation graphs. However, for large-scale industrial applications, MLPs remain the major workhorse, despite common (implicit) underlying graphs and suitability for GML formalisms. One reason for this academic-industrial gap is the challenges in scalability and deployment brought by data dependency in GNNs (Zhang et al., 2020; Jia et al., 2020), which makes GNNs hard to deploy for latency-constrained applications that require fast inference.

Neighborhood fetching caused by graph dependency is one of the major sources of GNN latency. Inference on a target node necessitates fetching topology and features of many neighbor nodes, especially on small-world graphs (detailed discussion in Section 4). Common inference acceleration techniques like pruning (Zhou et al., 2021) and quantization (Tailor et al., 2021; Zhao et al., 2020) can speed up GNNs to some extent by reducing Multiplication-and-ACcumulation (MAC) operations.

---

[*]Work done when author was an intern at Snap Inc. Code available at https://github.com/snap-research/graphless-neural-networks

However, their improvements are limited given the graph dependency is not resolved. Unlike GNNs, MLPs have no dependency on graph data and are easier to deploy than GNNs. They also enjoy the auxiliary benefit of sidestepping the cold-start problem that often happens during the online prediction of relational data (Wei et al., 2020), meaning MLPs can infer reasonably even when neighbor information of a new encountered node is not immediately available. On the other hand, this lack of graph dependency typically hurts for relational learning tasks, limiting MLP performance on GML tasks compared to GNNs. We thus ask: *can we bridge the two worlds, enjoying the low-latency, dependency-free nature of MLPs and the graph context-awareness of GNNs at the same time?*

**Present work.** Our key finding is that it is possible to distill knowledge from GNNs to MLPs without losing significant performance, but reducing the inference time *drastically* for node classification. The knowledge distillation (KD) can be done offline, coupled with model training. In other words, we can shift considerable work from the latency-constrained inference step, where time reduction in milliseconds makes a huge difference, to the less time-sensitive training step, where time cost in hours or days is often tolerable. We call our approach Graph-less Neural Network (GLNN). Specifically, GLNN is a modeling paradigm involving KD from a GNN teacher to a student MLP; the resulting GLNN is an MLP optimized through KD, so it enjoys the benefits of graph context-awareness in training but has no graph dependency in inference. Regarding speed, GLNNs have superior efficiency and are **146×-273×** faster than GNNs and **14×-27×** faster than other inference acceleration methods. Regarding performance, under a production setting involving both transductive and inductive predictions on 7 datasets, GLNN accuracies improve over MLPs by 12.36% on average and match GNNs on 6/7 datasets. We comprehensively study when and why GLNNs can achieve competitive results as GNNs. Our analysis suggests the critical factors for such great performance are large MLP sizes and high mutual information between node features and labels. Our observations align with recent results in vision and language, which posit that large enough (or slightly modified) MLPs can achieve similar results as CNNs and Transformers (Liu et al., 2021; Tolstikhin et al., 2021; Melas-Kyriazi, 2021; Touvron et al., 2021; Ding et al., 2021). Our core contributions are as follows:

- We propose GLNN, which eliminates neighbor-fetching latency in GNN inference via KD to MLP.
- We show GLNNs has competitive performance as GNNs, while enjoying **146×-273×** faster inference than vanilla GNNs and **14×-27×** faster inference than other inference acceleration methods.
- We study GLNN properties comprehensively by investigating their performance under different settings, how they work as regularizers, their inductive bias, expressiveness, and limitations.

## 2 RELATED WORK

**Graph Neural Networks.** The early GNNs generalize convolution nets to graphs (Bruna et al., 2014; Defferrard et al., 2017) and later simplified to message-passing neural net (MPNN) by GCN (Kipf & Welling, 2016). Most GNNs after can be put as MPNNs. For example, GAT employs attention (Veličković et al., 2017), PPNP employs personalized PageRank (Klicpera et al., 2019), GCNII and DeeperGCN employ residual connections and dense connections (Chen et al., 2020; Li et al., 2019).

**Inference Acceleration.** Inference acceleration have been proposed by hardware improvements (Chen et al., 2016; Judd et al., 2016) and algorithmic improvements through pruning (Han et al., 2015), quantization (Gupta et al., 2015). For GNNs, pruning (Zhou et al., 2021) and quantizing GNN parameters (Zhao et al., 2020) have been studied. These approaches speed up GNN inference to a certain extent but do not eliminate the neighbor-fetching latency. In contrast, our cross-model KD solves this issue. Concurrently, Graph-MLP also tries to bypass GNN neighbor fetching (Hu et al., 2021) by training an MLP with a neighbor contrastive loss, but it only considers transductive but not the more practical inductive setting. Some sampling works focus on speed up GNN training (Zou et al., 2019; Chen et al., 2018), which are complementary to our goal on inference acceleration.

**GNN distillation.** Existing GNN KD works try to distill large GNNs to smaller GNNs. LSP (Yang et al., 2021b) and TinyGNN (Yan et al., 2020) do KD while preserving local information. Their students are GNNs with fewer parameters but not necessarily fewer layers. Thus, both designs still require latency-inducing fetching. GFKD (Deng & Zhang, 2021) does graph-level KD via graph generation. In GFKD, data instances are independent graphs, whereas we focus on dependent nodes within a graph. GraphSAIL (Xu et al., 2020) uses KD to learn students work well on new data while preserving performance on old data. CPF (Yang et al., 2021a) combines KD and label propagation (LP). The student in CPF is not a GNN, but it is still heavily graph-dependent as it uses LP.

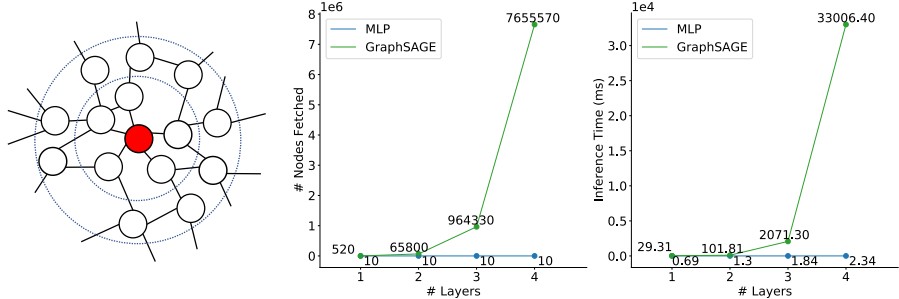

Figure 1: The number of fetches and the inference time of GNNs are both magnitudes more than MLPs and grow exponentially as functions of the number of layers. **Left**: neighbors need to be fetched for two GNN layers. **Middle**: the total number of fetches for inference. **Right**: the total inference time. (Inductive inference for 10 random nodes on OGB `Products` (Hu et al., 2020))

## 3 PRELIMINARIES

**Notations.** For GML tasks, the input is usually a graph and its node features, which we write as $\mathcal{G} = (\mathcal{V}, \mathcal{E})$, with $\mathcal{V}$ stands for all nodes, and $\mathcal{E}$ stands for all edges. Let $N$ denote the total number of nodes. We use $\boldsymbol{X} \in \mathbb{R}^{N \times D}$ to represent node features, with row $\boldsymbol{x}_v$ being the $D$-dimensional feature of node $v \in \mathcal{V}$. We represent edges with an adjacency matrix $\boldsymbol{A}$, with $A_{u,v} = 1$ if edge $(u, v) \in \mathcal{E}$, and 0 otherwise. For node classification, one of the most important GML applications, the prediction targets are $\boldsymbol{Y} \in \mathbb{R}^{N \times K}$, where row $\boldsymbol{y}_v$ is a $K$-dim one-hot vector for node $v$. For a given $\mathcal{G}$, usually a small portion of nodes will be labeled, which we mark using superscript $^L$, i.e. $\mathcal{V}^L$, $\boldsymbol{X}^L$, and $\boldsymbol{Y}^L$. The majority of nodes will be unlabeled, and we mark using the superscript $^U$, i.e. $\mathcal{V}^U$, $\boldsymbol{X}^U$, and $\boldsymbol{Y}^U$.

**Graph Neural Networks.** Most GNNs fit under the message-passing framework, where the representation $\boldsymbol{h}_v$ of each node $v$ is updated iteratively in each layer by collecting messages from its neighbors denoted as $\mathcal{N}(v)$. For the $l$-th layer, $\boldsymbol{h}_v^{(l)}$ is obtained from the previous layer representation $\boldsymbol{h}_u^{(l-1)}$ ($\boldsymbol{h}_u^{(0)} = \boldsymbol{x}_u$) via an aggregation operation AGGR followed by an UPDATE operation as

$$\boldsymbol{h}_{N(v)}^{(l)} = \text{AGGR}(\{\boldsymbol{h}_u^{(l-1)} : u \in \mathcal{N}(v)\}) \qquad \text{and} \qquad \boldsymbol{h}_v^{(l)} = \text{UPDATE}(\boldsymbol{h}_{N(v)}^{(l)}, \boldsymbol{h}_v^{(l-1)})$$

## 4 MOTIVATION

GNNs have considerable inference latency due to graph dependency. One more GNN layer means fetching one more hop of neighbors. To infer a node with a $L$-layer GNN on a graph with average degree $R$ requires $\mathcal{O}(R^L)$ fetches. $R$ can be large for real-world graphs, e.g. 208 for the Twitter (Ching et al., 2015). Also, as layer fetching must be done sequentially, the total latency explodes quickly as $L$ increases. Figure 1 shows the dependency added by each GNN layer and the exponential explosion of inference time. In contrast, the MLP inference time is much smaller and grows linearly. This marked gap contributes greatly to the practicality of MLPs in industrial applications over GNNs.

The node-fetching latency is exacerbated by two factors: firstly, newer GNN architectures are getting deeper from 64 layers (Chen et al., 2020) to even 1001 layers (Li et al., 2021). Secondly, industrial-scale graphs are frequently too large to fit into the memory of a single machine (Jin et al., 2022), necessitating sharding of the graph out of the main memory. For example, Twitter has 288M monthly active users (nodes) and an estimated 60B followers (edges) as of 3/2015. Facebook has 1.39B active users with more than 400B edges as of 12/2014 (Ching et al., 2015). Even when stored in a sparse-matrix-friendly format (often COO or CSR), these graphs are on the order of TBs and are constantly growing. Moving away from in-memory storage results in even slower neighbor-fetching.

MLPs, on the other hand, lack the means to exploit graph topology, which hurts their performance for node classification. For example, test accuracy on `Products` is 78.61 for GraphSAGE compared to 62.47 for an equal-sized MLP. Nonetheless, recent results in vision and language posit that large (or slightly modified) MLPs can achieve similar results as CNNs and Transformers (Liu et al., 2021). We thus also ask: Can we bridge the best of GNNs and MLPs to get high-accuracy and low-latency models? This motivates us to do cross-model KD from GNNs to MLPs.

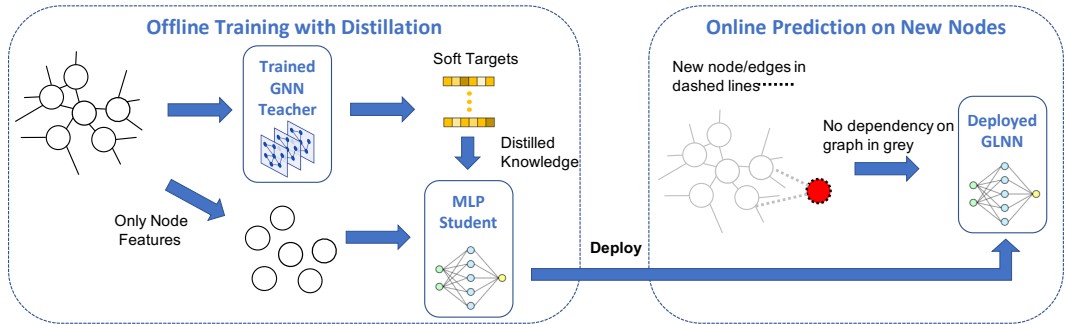

Figure 2: The GLNN framework: In offline training, a trained GNN teacher is applied on the graph for soft targets. Then, a student MLP is trained on node features guided by the soft targets. The distilled MLP, now GLNN, is deployed for online predictions. Since graph dependency is eliminated for inference, GLNNs infer much faster than GNNs, and hence the name "Graph-less Neural Network."

## 5 GRAPH-LESS NEURAL NETWORKS

We introduce GLNN and answer exploration questions of its properties: **1)** How do GLNNs compare to MLPs and GNNs? **2)** Can GLNNs work well under both transductive and inductive settings? **3)** How do GLNNs compare to other inference acceleration methods? **4)** How do GLNNs benefit from KD? **5)** Do GLNNs have sufficient model expressiveness? **6)** When will GLNNs fail to work?

### 5.1 THE GLNN FRAMEWORK

The idea of GLNN is straightforward, yet as we will see, extremely effective. In short, we train a "boosted" MLP via KD from a teacher GNN. KD was introduced in Hinton et al. (2015), where knowledge was transferred from a cumbersome teacher to a simpler student. In our case, we generate soft targets $z_v$ for each node $v$ with a teacher GNN. Then we train a student MLP with both true labels $y_v$ and $z_v$. The objective is as Equation 1, with $\lambda$ being a weight parameter, $\mathcal{L}_{label}$ being the cross-entropy between $y_v$ and student predictions $\hat{y}_v$, $\mathcal{L}_{teacher}$ being the KL-divergence.

$$\mathcal{L} = \lambda\Sigma_{v\in\mathcal{V}^L}\mathcal{L}_{label}(\hat{y}_v, y_v) + (1-\lambda)\Sigma_{v\in\mathcal{V}}\mathcal{L}_{teacher}(\hat{y}_v, z_v) \tag{1}$$

The model after KD, i.e. GLNN, is essentially a MLP. Therefore, GLNNs have no graph dependency during inference and are as fast as MLPs. On the other hand, through offline KD, GLNN parameters are optimized to predict and generalize as well as GNNs, with the added benefit of faster inference and easier deployment. In Figure 2, we show the offline KD and online inference steps of GLNNs.

### 5.2 EXPERIMENT SETTINGS

**Datasets.** We consider all five datasets used in the CPF paper (Yang et al., 2021a), i.e. `Cora`, `Citeseer`, `Pubmed`, `A-computer`, and `A-photo`. To fully evaluate our method, we also include two more larger OGB datasets (Hu et al., 2020), i.e. `Arxiv` and `Products`.

**Model Architectures.** For consistent results, we use GraphSAGE (Hamilton et al., 2017) with GCN aggregation as the teacher. We conduct ablation studies of other GNN teachers like GCN (Kipf & Welling, 2016), GAT (Veličković et al., 2017) and, APPNP (Klicpera et al., 2019) in Section 6.

**Evaluation Protocol.** For all experiments in this section, we report the average and standard deviation over ten runs with different random seeds. Model performance is measured as accuracy, and results are reported on test data with the best model selected using validation data.

**Transductive vs. Inductive.** Given $\mathcal{G}$, $X$, and $Y^L$, we consider node classification under two settings: transductive (*tran*) and inductive (*ind*). For *ind*, we hold out some test data for inductive evaluation only. We first select inductive nodes $\mathcal{V}^U_{ind} \subset \mathcal{V}^U$, which partitions $\mathcal{V}^U$ into the disjoint inductive subset and observed subset, i.e. $\mathcal{V}^U = \mathcal{V}^U_{obs} \sqcup \mathcal{V}^U_{ind}$. Then we hold out $v \in \mathcal{V}^U_{ind}$ and all edges connected to $v \in \mathcal{V}^U_{ind}$, which leads to two disjoint graphs $\mathcal{G} = \mathcal{G}_{obs} \sqcup \mathcal{G}_{ind}$ with no shared nodes or

Table 1: GLNNs outperform MLPs by large margins and match GNNs on 5 of 7 datasets under the **transductive** setting. $\Delta_{MLP}$ ($\Delta_{GNN}$) represents difference between the GLNN and a trained MLP (GNN). Results show accuracy (higher is better); $\Delta_{GNN} \geq 0$ indicates GLNN outperforms GNN.

| Datasets | SAGE | MLP | GLNN | $\Delta_{MLP}$ | $\Delta_{GNN}$ |
|---|---|---|---|---|---|
| Cora | $80.52 \pm 1.77$ | $59.22 \pm 1.31$ | $\mathbf{80.54 \pm 1.35}$ | 21.32 (36.00%) | 0.02 (0.02%) |
| Citeseer | $70.33 \pm 1.97$ | $59.61 \pm 2.88$ | $\mathbf{71.77 \pm 2.01}$ | 12.16 (20.40%) | 1.44 (2.05%) |
| Pubmed | $75.39 \pm 2.09$ | $67.55 \pm 2.31$ | $\mathbf{75.42 \pm 2.31}$ | 7.87 (11.65%) | 0.03 (0.04%) |
| A-computer | $82.97 \pm 2.16$ | $67.80 \pm 1.06$ | $\mathbf{83.03 \pm 1.87}$ | 15.23 (22.46%) | 0.06 (0.07%) |
| A-photo | $90.90 \pm 0.84$ | $78.77 \pm 1.74$ | $\mathbf{92.11 \pm 1.08}$ | 13.34 (16.94%) | 1.21 (1.33%) |
| Arxiv | $\mathbf{70.92 \pm 0.17}$ | $56.05 \pm 0.46$ | $63.46 \pm 0.45$ | 7.41 (13.24%) | -7.46 (-10.52%) |
| Products | $\mathbf{78.61 \pm 0.49}$ | $62.47 \pm 0.10$ | $68.86 \pm 0.46$ | 6.39 (10.23%) | -9.75 (-12.4%) |

Table 2: Enlarged GLNNs match the performance of GNNs on the OGB datasets. For Arxiv, we use MLPw4 (GLNNw4). For Products, we use MLPw8 (GLNNw8).

| Datasets | SAGE | MLP+ | GLNN+ | $\Delta_{MLP}$ | $\Delta_{GNN}$ |
|---|---|---|---|---|---|
| Arxiv | $70.92 \pm 0.17$ | $55.31 \pm 0.47$ | $\mathbf{72.15 \pm 0.27}$ | 16.85 (30.46%) | 0.51 (0.71%) |
| Products | $\mathbf{78.61 \pm 0.49}$ | $64.50 \pm 0.45$ | $77.65 \pm 0.48$ | 13.14 (20.38%) | -0.97 (-1.23%) |

edges. Node features and labels are partitioned into three disjoint sets, i.e. $\boldsymbol{X} = \boldsymbol{X}^L \sqcup \boldsymbol{X}^U_{obs} \sqcup \boldsymbol{X}^U_{ind}$, and $\boldsymbol{Y} = \boldsymbol{Y}^L \sqcup \boldsymbol{Y}^U_{obs} \sqcup \boldsymbol{Y}^U_{ind}$. Concretely, the input/output of both settings become:

- *tran*: train on $\mathcal{G}$, $\boldsymbol{X}$, and $\boldsymbol{Y}^L$; evaluate on $(\boldsymbol{X}^U, \boldsymbol{Y}^U)$; KD uses $\boldsymbol{z}_v$ for $v \in \mathcal{V}$.
- *ind*: train on $\mathcal{G}_{obs}$, $\boldsymbol{X}^L$, $\boldsymbol{X}^U_{obs}$, and $\boldsymbol{Y}^L$; evaluate on $(\boldsymbol{X}^U_{ind}, \boldsymbol{Y}^U_{ind})$; KD uses $\boldsymbol{z}_v$ for $v \in \mathcal{V}^L \sqcup \mathcal{V}^U_{obs}$.

Note that for *tran*, all the nodes in the graph including the validation and test nodes are used to generate $\boldsymbol{z}$. A discussion of this choice along with other experiment details are in Appendix A.

## 5.3 How do GLNNs compare to MLPs and GNNs?

We start by comparing GLNNs to MLPs and GNNs with the same number of layers and hidden dimensions. We first consider the standard transductive setting, so our results in Table 1 are directly comparable to results reported in previous literature like Yang et al. (2021a) and Hu et al. (2020).

As shown in Table 1, the performance of all GLNNs improve over MLPs by large margins. On smaller datasets (first 5 rows), GLNNs can even outperform the teacher GNNs. In other words, for each task, with the same parameter budget, there exists a set of MLP parameters that has GNN-competitive performance (detailed discussion in Sections 5.6 and 5.7). For the larger OGB datasets (last 2 rows), the GLNN performance is improved over MLPs but still worse than the teacher GNNs. However, as we show in Table 2, this gap can be mitigated by increasing MLP size to MLPw$i$[1]. In Figure 3 (right), we visualize the trade-off between prediction accuracy and model inference time with different model sizes. We show that gradually increasing GLNN size pushes its performance to be close to SAGE. On the other hand, when we reduce the number of layers of SAGE[2], the accuracy quickly drops to be worse than GLNNs. A detailed discussion of the rationale for increasing MLP sizes is in Appendix B.

## 5.4 Can GLNNs work well under both transductive and inductive settings?

Although transductive is the commonly studied setting for node classification, it does not encompass prediction on unseen nodes. Therefore, it may not be the best way to evaluate a deployed model, which must often generate predictions for new data points as well as reliably maintain performance on old ones. Thus, to better understand the effectiveness of GLNN, we also consider their performance under a realistic production setting, which contains both transductive and inductive predictions.

To evaluate a model inductively, we hold out some test nodes from training to form an inductive set, i.e. $\mathcal{V}^U = \mathcal{V}^U_{obs} \sqcup \mathcal{V}^U_{ind}$. In production, a model might be re-trained periodically, e.g. weekly. The hold-out nodes in $V^U_{ind}$ represent new nodes entered the graph between two trainings. $V^U_{ind}$ is usually

---

[1] -w$i$ means $i$-times wider hidden layers, e.g. hidden layers of MLPw4 are 4-times wider than the given MLP.

[2] -L$i$ is used to explicitly note a model with $i$ layers, e.g. SAGE-L2 represents a 2-layer SAGE.

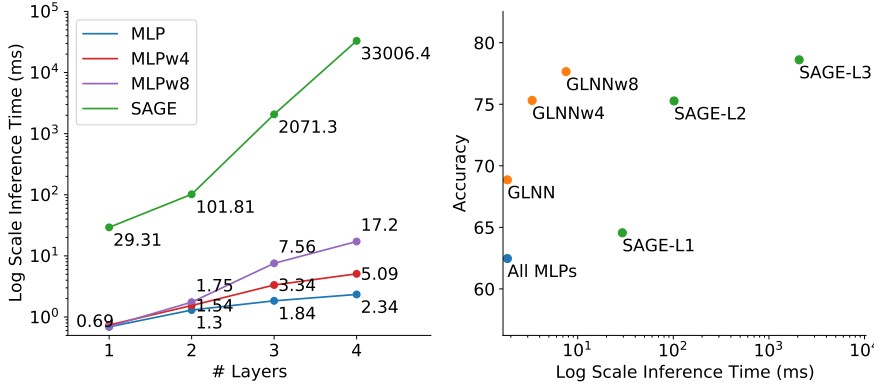

Figure 3: Enlarged MLPs (GLNNs) can match GNN accuracy, but infer dramatically faster. Plots are under the same setting as Figure 1. **Left**: inference time of MLPs vs. GNN (SAGE) for different model sizes. **Right**: model accuracy vs. inference time. Note: time axes are log-scaled.

Table 3: GLNNs match GNN performance on a production setting with both **inductive** and **transductive** predictions. We use MLP for the 5 CPF datasets, MLPw4 for `Arxiv`, and MLPw8 for `Products`. *ind* results on $V_{ind}^U$, *tran* results on $V_{obs}^U$, and the interpolated *prod* results are reported.

| Datasets | Eval | SAGE | MLP/MLP+ | GLNN/GLNN+ | $\Delta_{MLP}$ | $\Delta_{GNN}$ |
|---|---|---|---|---|---|---|
| Cora | *prod* | **79.29** | 58.98 | 78.28 | 19.30 (32.72%) | -1.01 (-1.28%) |
| | *ind* | $81.33 \pm 2.19$ | $59.09 \pm 2.96$ | $73.82 \pm 1.93$ | 14.73 (24.93%) | -7.51 (-9.23%) |
| | *tran* | $78.78 \pm 1.92$ | $58.95 \pm 1.66$ | $79.39 \pm 1.64$ | 20.44 (34.66%) | 0.61 (0.77%) |
| Citeseer | *prod* | 68.38 | 59.81 | **69.27** | 9.46 (15.82%) | 0.89 (1.30%) |
| | *ind* | $69.75 \pm 3.59$ | $60.06 \pm 5.00$ | $69.25 \pm 2.25$ | 9.19 (15.30%) | -0.5 (-0.7%) |
| | *tran* | $68.04 \pm 3.34$ | $59.75 \pm 2.48$ | $69.28 \pm 3.12$ | 9.63 (15.93%) | 1.24 (1.82%) |
| Pubmed | *prod* | **74.88** | 66.80 | 74.71 | 7.91 (11.83%) | -0.17 (-0.22%) |
| | *ind* | $75.26 \pm 2.57$ | $66.85 \pm 2.96$ | $74.30 \pm 2.61$ | 7.45 (11.83%) | -0.96 (-1.27%) |
| | *tran* | $74.78 \pm 2.22$ | $66.79 \pm 2.90$ | $74.81 \pm 2.39$ | 8.02 (12.01%) | 0.03 (0.04%) |
| A-computer | *prod* | 82.14 | 67.38 | **82.29** | 14.90 (22.12%) | 0.15 (0.19%) |
| | *ind* | $82.08 \pm 1.79$ | $67.84 \pm 1.78$ | $80.92 \pm 1.36$ | 13.08 (19.28%) | -1.16 (-1.41%) |
| | *tran* | $82.15 \pm 1.55$ | $67.27 \pm 1.36$ | $82.63 \pm 1.40$ | 15.36 (22.79%) | 0.48 (0.58%) |
| A-photo | *prod* | 91.08 | 79.25 | **92.38** | 13.13 (16.57%) | 1.30 (1.42%) |
| | *ind* | $91.50 \pm 0.79$ | $79.44 \pm 1.72$ | $91.18 \pm 0.81$ | 11.74 (14.78%) | -0.32 (-0.35%) |
| | *tran* | $90.80 \pm 0.77$ | $79.20 \pm 1.64$ | $92.68 \pm 0.56$ | 13.48 (17.01%) | 1.70 (1.87%) |
| Arxiv | *prod* | **70.73** | 55.30 | 65.09 | 9.79 (17.70%) | -5.64 (-7.97%) |
| | *ind* | $70.64 \pm 0.67$ | $55.40 \pm 0.56$ | $60.48 \pm 0.46$ | 4.3 (7.76%) | -10.94 (-15.49%) |
| | *tran* | $70.75 \pm 0.27$ | $55.28 \pm 0.49$ | $71.46 \pm 0.33$ | 11.16 (20.18%) | -4.31 (-6.09%) |
| Products | *prod* | **76.60** | 63.72 | 75.77 | 12.05 (18.91%) | -0.83 (-1.09%) |
| | *ind* | $76.89 \pm 0.53$ | $63.70 \pm 0.66$ | $75.16 \pm 0.34$ | 11.44 (17.96%) | -1.73 (-2.25%) |
| | *tran* | $76.53 \pm 0.55$ | $63.73 \pm 0.69$ | $75.92 \pm 0.61$ | 12.20 (19.15%) | -0.61 (-0.79%) |

small compared to $V_{obs}^U$ – e.g. Graham (2012) estimates 5-7% for the fastest-growing tech startups. In our case, to mitigate randomness and better evaluate generalizability, we use $V_{ind}^U$ containing 20% of the test data. We also evaluate on $V_{obs}^U$ containing the other 80% of the test data, representing the standard transductive prediction on observed unlabeled nodes, since inference is commonly redone on existing nodes in real-world cases. We report both results and a interpolated production (*prod*) results in Table 3. The *prod* results paint a clearer picture of model generalization as well as accuracy in production. See Section 6 for an ablation study of different inductive split rates other than 20-80.

In Table 3, we see that GLNNs can still improve over MLP by large margins for inductive predictions. On 6/7 datasets, the GLNN *prod* performance are competitive to GNNs, which supports deploying GLNN as a much faster model with no or only slight performance loss. On the `Arxiv` dataset, the GLNN performance is notably less than GNNs – we hypothesize this is due to `Arxiv` having a particularly challenging data split which causes distribution shift between test nodes and training

Table 4: While other inference acceleration methods speed up SAGE, they are considerably slower than GLNNs. Numbers (in *ms*) are inductive inference time on 10 randomly chosen nodes.

| Datasets | SAGE | QSAGE | PSAGE | Neighbor Sample | GLNN+ |
|---|---|---|---|---|---|
| Arxiv | 489.49 | 433.90 (1.13×) | 465.43 (1.05×) | 91.03 (5.37×) | **3.34 (146.55×)** |
| Products | 2071.30 | 1946.49 (1.06×) | 2001.46 (1.04×) | 107.71 (19.23×) | **7.56 (273.98×)** |

nodes, which is hard for GLNNs to capture without utilizing neighbor information like GNNs. However, we note that GLNN performance is substantially improved over MLP.

## 5.5 HOW DO GLNNs COMPARE TO OTHER INFERENCE ACCELERATION METHODS?

Common techniques of inference acceleration include pruning and quantization. These approaches can reduce model parameters and Multiplication-and-ACcumulation (MACs) operations. Still, they don't eliminate neighbor-fetching latency. Therefore, their speed gain on GNNs is less significant than on NNs. For GNNs, neighbor sampling is also used to reduce the fetching latency. We show an explicit speed comparison between vanilla SAGE, quantized SAGE from FP32 to INT8 (QSAGE), SAGE with 50% weights pruned (PSAGE), inference neighbor sampling with fan-out 15, and GLNN in Table 4. With the same setting as Figure 1, we see that GLNN is considerably faster.

Two other kinds of methods considered as inference acceleration are GNN-to-GNN KD like TinyGNN (Yan et al., 2020) and Graph Augmented-MLPs (GA-MLPs) like SGC (Wu et al., 2019) or SIGN (Frasca et al., 2020). Inference of GNN-to-GNN KD is likely to be slower than a GNN-L$i$ with the same $i$ as the student, since there will usually be some extra overheads like the Peer-Aware Module (PAM) in TinyGNN. GA-MLPs precompute augmented node features and apply MLPs to them. With precomputation, their inference time will be the same as MLPs for dimension-preserving augmentation (SGC) and the same as enlarged MLPw$i$ for augmentation involves concatenation (SIGN). Thus, for both kinds of approaches, it is sufficient to compare GLNN with GNN-L$i$ and MLPw$i$, which we have already shown in Figure 3 (left). We see that GNN-L$i$s are much slower than MLPs. For GA-MLPs, since full pre-computation cannot be done for inductive nodes, GA-MLPs still need to fetch neighbor nodes. This makes them much slower than MLPw$i$ in the inductive setting, and even slower than pruned GNNs and TinyGNN as shown in Zhou et al. (2021).

## 5.6 HOW DOES GLNN BENEFIT FROM DISTILLATION?

We showed that GNNs are markedly better than MLPs on node classification tasks. But, with KD, GLNNs can often become competitive to GNNs. This indicates that there exist suitable MLP parameters which can well approximate the ideal prediction function from node features to labels. However, these parameters can be difficult to learn through standard stochastic gradient descent. We hypothesize that KD helps to find them through regularization and transfer of inductive bias.

First, we show that KD can help to regularize the student model. From loss curves of a directly trained MLP and the GLNN in Figure 4, we see the gap between training and validation loss is visibly larger for MLPs than GLNNs, and MLPs show obvious overfitting trends. Second, we analyze the inductive bias that makes GNNs powerful on node classification, which suggests that node inferences should be influenced by the graph topology. Whereas MLPs have less inductive bias. Similar difference exists between Transformers (Vaswani et al., 2017) and MLPs. Liu et al. (2021) shows that the inductive bias in Transformers can be mitigated by a simple gate on large MLPs. For node classification, we hypothesize that KD helps to mitigate the inductive bias, so GLNNs can perform competitively. Soft labels from GNN teachers are heavily influenced by the graph topology due to inductive bias. They maintain nonzero probabilities on classes other than the ground truth provided by labels, which can be useful for the student to learn to complement the missing inductive bias in MLPs. To evaluate this hypothesis quantitatively, we define the cut loss $\mathcal{L}_{cut} \in [0, 1]$ in Equation 2 to measure the consistency between model predictions and graph topology (details in Appendix C):

$$\mathcal{L}_{cut} = \frac{Tr(\hat{\boldsymbol{Y}}^T \boldsymbol{A} \hat{\boldsymbol{Y}})}{Tr(\hat{\boldsymbol{Y}}^T \boldsymbol{D} \hat{\boldsymbol{Y}})} \tag{2}$$

Here $\hat{\boldsymbol{Y}} \in [0, 1]^{N \times K}$ is the soft classification probability output by the model, $\boldsymbol{A}$ and $\boldsymbol{D}$ are the adjacency and degree matrices. When $\mathcal{L}_{cut}$ is close to 1, it means the predictions and the graph

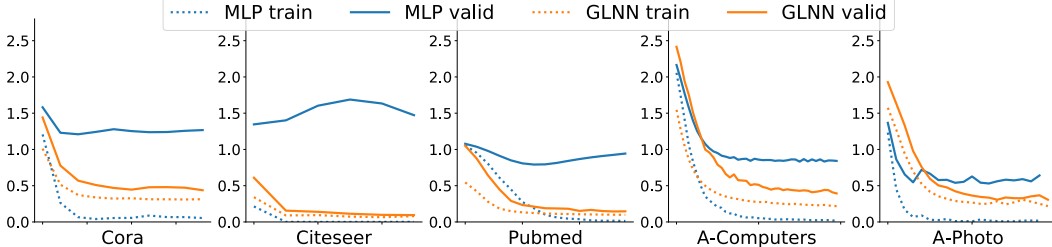

Figure 4: Loss curves on CPF datasets show GLNN distillation can help to regularize the training. Here the training loss of GLNN is on hard labels, only corresponding to the first term in Equation 1.

topology are very consistent. In our experiment, we observe that the average $\mathcal{L}_{cut}$ for SAGE over five CPF datasets is 0.9221, which means high consistency. The same $\mathcal{L}_{cut}$ for MLPs is only 0.7644, but for GLNNs it is 0.8986. This shows that the GLNN predictions indeed benefit from the graph topology knowledge contained in the teacher outputs (the full table of $\mathcal{L}_{cut}$ values in Appendix C).

## 5.7 DO GLNNS HAVE ENOUGH MODEL EXPRESSIVENESS?

Intuitively, the addition of neighbor information makes GNNs more powerful than MLPs when classifying nodes. Thus, a natural question regarding KD from GNNs to MLPs is whether MLPs are expressive enough to represent graph data as well as GNNs. Many recent works studied GNN model expressiveness (Xu et al., 2018; Chen et al., 2021). The latter analyzed GNNs and GA-MLPs for node classification and characterized expressiveness as the number of equivalence classes of rooted graphs induced by the model (formal definitions in Appendix D). The conclusion is that GNNs are more powerful than GA-MLPs, but in most real-world cases their expressiveness is indistinguishable.

We adopt the analysis framework from Chen et al. (2021) and show in Appendix D that the number of equivalence classes induced by GNNs and MLPs are $\binom{|\mathcal{X}|+m-2}{m-1}^{2^L-1}$ and $|\mathcal{X}|$ respectively. Here $m$ denotes the max node degree, $L$ denotes the number of GNN layers, and $\mathcal{X}$ denotes the set of all possible node features. The former is apparently larger which concludes that GNNs are more expressive. Empirically, however, the gap makes little difference when $|\mathcal{X}|$ is large. In real applications, node features can be high dimensional like bag-of-words, or even word embeddings, thus making $|\mathcal{X}|$ enormous. Like for bag-of-words, $|\mathcal{X}|$ is in the order of $\mathcal{O}(p^D)$, where $D$ is the vocabulary size, and $p$ is the max word frequency. The expressiveness of a L-layer GNN is lower bounded by $\binom{|\mathcal{X}|+m-2}{m-1}^{2^L-1} = \mathcal{O}(p^{D(m-1)(2^L-1)})$, but empirically, both MLPs and GNNs should have enough expressiveness given $D$ is usually hundreds or bigger (see Table 5).

## 5.8 WHEN WILL GLNNS FAIL TO WORK?

As discussed in Section 5.7 and Appendix D, the goal of GML node classification is to fit a function $f$ on the rooted graph $\mathcal{G}^{[i]}$ and label $\boldsymbol{y}_i$ . From the information theoretic perspective, fitting $f$ by minimizing the commonly used cross-entropy loss is equivalent to maximizing the mutual information (MI), $I(\mathcal{G}^{[i]}; \boldsymbol{y}_i)$ as shown in Qin et al. (2020). If we consider $\mathcal{G}^{[i]}$ as a joint distribution of two random variables $\boldsymbol{X}^{[i]}$ and $\mathcal{E}^{[i]}$ representing the node features and edges in $\mathcal{G}^{[i]}$ respectively, we have

$$I(\mathcal{G}^{[i]}; \boldsymbol{y}_i) = I(\boldsymbol{X}^{[i]}, \mathcal{E}^{[i]}; \boldsymbol{y}_i) = I(\mathcal{E}^{[i]}; \boldsymbol{y}_i) + I(\boldsymbol{X}^{[i]}; \boldsymbol{y}_i | \mathcal{E}^{[i]}) \qquad (3)$$

$I(\mathcal{E}^{[i]}; \boldsymbol{y}_i)$ only depends on edges and labels, thus MLPs can only maximize $I(\boldsymbol{X}^{[i]}; \boldsymbol{y}_i | \mathcal{E}^{[i]})$. In the extreme case, $I(\boldsymbol{X}^{[i]}; \boldsymbol{y}_i | \mathcal{E}^{[i]})$ can be zero when $\boldsymbol{y}^{[i]}$ is conditionally independent from $\boldsymbol{X}^{[i]}$ given $\mathcal{E}^{[i]}$. For example, when every node is labeled by its degree or whether it forms a triangle. Then MLPs won't be able to fit meaningful functions, and neither will GLNNs. However, such cases are typically rare, and unexpected in practical settings our work is mainly concerned with. For real GML tasks, node features and structural roles are often highly correlated (Lerique et al., 2020), hence MLPs can achieve reasonable results even only based on node features, and thus GLNNs can potentially achieve much better results. We study the failure case of GLNNs by creating a low MI scenario in Section 6.

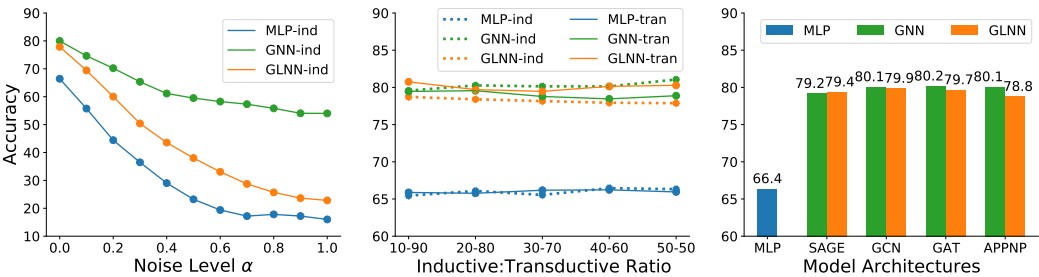

Figure 5: **Left**: Node feature noise. GLNN has comparable performance to GNNs only when nodes are less noisy. Adding more noise decreases GLNN performance faster than GNNs. **Middle**: Inductive split rate. Altering the inductive:transductive ratio in the production setting doesn't affect the accuracy much. **Right**: Teacher GNN architecture. GLNNs can learn from different GNN teachers to improve over MLPs and achieve comparable results. Accuracies are averaged over five CPF datasets.

## 6 ABLATION STUDIES

In this section, we do ablation studies of GLNNs on node feature noise, inductive split rates, and teacher GNN architecture. Reported results are test accuracies averaged over five datasets in CPF. More experiments can be found in Appendix including advanced GNN teachers (Appendix F), GA-MLP student (Appendix G), and non-homogeneous data (Appendix I).

**Noisy node features.** Following Section 5.8, we investigate failure cases of GLNN by adding different levels of Gaussian noise to node features to decrease their mutual information with labels. Specifically, we replace $X$ with $\tilde{X} = (1 - \alpha)X + \alpha\epsilon$. $\epsilon$ is an isotropic Gaussian independent from $X$, and $\alpha \in [0, 1]$ denotes the noise level. We show the inductive performance of MLP, GNN, and GLNN under different noise levels in Figure 5 (left). We see that as $\alpha$ increases, the accuracy of MLPs and GLNNs decrease faster than GNNs, while the performance of GLNNs and GNNs are still comparable for small $\alpha$s. When $\alpha$ reaches 1, $\tilde{X}$ and $Y$ will become independent corresponding to the extreme case discussed in Section 5.8. A more detailed discussion is in Appendix J.

**Inductive split rate.** In Section 5.4, we use a 20-80 split of the test data for inductive evaluation. In Figure 5 (middle), we show the results under different split rates (More detailed plots in Appendix H). We see that as the inductive portion increase, GNN and MLP performance stays roughly the same, and the GLNN inductive performance drops slightly. We only consider rates up to 50-50 since having 50% or even more inductive nodes is highly atypical in practice. When a large amount of new data are encountered, practitioners can opt to retrain the model on all the data before deployment.

**Teacher GNN architecture.** We used SAGE to represent GNNs so far. In Figure 5 (right), we show results with other various GNN teachers, e.g. GCN, GAT, and APPNP. We see that GLNNs can learn from different teachers and improve over MLPs. The performance is similar for all four teachers, with the GLNN distilled from APPNP very slightly worse than others. In fact, a similar phenomenon has been observed in Yang et al. (2021a) as well, i.e. APPNP benefits the student the least. One possible reason is that the first step of APPNP is to utilize the node's own feature for prediction (prior to propagating over the graph), which is very similar to what the student MLP is doing, and thus provides less additional information to MLPs than other teachers.

## 7 CONCLUSION AND FUTURE WORK

In this paper, we explored whether we can bridge the best of GNNs and MLPs to achieve accurate and fast GML models for deployment. We found that KD from GNNs to MLPs helps to eliminate inference graph dependency, which results in GLNNs that are 146×-273× faster than GNNs while enjoying competitive performance. We do a comprehensive study of GLNN properties. The promising results on 7 datasets across different domains show that GLNNs can be a handy choice for deploying latency-constraint models. In our experiments, the current version of GLNNs on the `Arxiv` dataset doesn't show competitive inductive performance. More advanced distillation techniques can potentially improve the GLNN performance, and we leave this investigation as future work.

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

# A  DETAILED EXPERIMENT SETTINGS

## A.1  DATASETS

Here we provide a detailed description of the datasets we used to support our argument. Out of these datasets, 4 of them are citation graphs. Cora, Citeseer, Pubmed, ogbn-arxiv with the node features being descriptions of the papers, either bag-of-word vector, TF-IDF vector, or word embedding vectors.

In Table 5, we provided the basic statistics of these datasets.

Table 5: Dataset Statistics.

| Dataset | # Nodes | # Edges | # Features | # Classes |
|---------|---------|---------|------------|-----------|
| Cora | 2,485 | 5,069 | 1,433 | 7 |
| Citeseer | 2,110 | 3,668 | 3,703 | 6 |
| Pubmed | 19,717 | 44,324 | 500 | 3 |
| A-computer | 13,381 | 245,778 | 767 | 10 |
| A-photo | 7,487 | 119,043 | 745 | 8 |
| Arxiv | 169,343 | 1,166,243 | 128 | 40 |
| Products | 2,449,029 | 61,859,140 | 100 | 47 |

For all datasets, we follow the setting in the original paper to split the data. Specifically, for the five smaller datasets from the CPF paper, we use the CPF splitting strategy and each random seed corresponds to a different split. For the OGB datasets, we follow the OGB official splits based on time and popularity for Arxiv and Products respectively.

## A.2  MODEL HYPERPARAMETERS

The hyperparameters of GNN models on each dataset are taken from the best hyperparameters provided by the CPF paper and the OGB official examples. For the student MLPs and GLNN s, unless otherwise specified with -w$i$ or -L$i$, we set the number of layers and the hidden dimension of each layer to be the same as the teacher GNN, so their total number of parameters stays the same as the teacher GNN.

Table 6: Hyperparameters for GNNs on five datasets from the CPF paper.

|  | SAGE | GCN | GAT | APPNP |
|--|------|-----|-----|-------|
| # layers | 2 | 2 | 2 | 2 |
| hidden dim | 128 | 64 | 64 | 64 |
| learning rate | 0.01 | 0.01 | 0.01 | 0.01 |
| weight decay | 0.0005 | 0.001 | 0.01 | 0.01 |
| dropout | 0 | 0.8 | 0.6 | 0.5 |
| fan out | 5,5 | - | - | - |
| attention heads | - | - | 8 | - |
| power iterations | - | - | - | 10 |

Table 7: Hyperparameters for GraphSAGE on OGB datasets.

| Dataset | Arxiv | Products |
|---------|-------|----------|
| # layers | 3 | 3 |
| hidden dim | 256 | 256 |
| learning rate | 0.01 | 0.003 |
| weight decay | 0 | 0 |
| dropout | 0.2 | 0.5 |
| normalization | batch | batch |
| fan out | [5, 10, 15] | [5, 10, 15] |

For GLNN s we do a hyperparameter search of learning rate from [0.01, 0.005, 0.001], weight decay from [0, 0.001, 0.002, 0.005, 0.01], and dropout from [0, 0.1, 0.2, 0.3, 0.4, 0.5, 0.6]

### A.3 Knowledge Distillation

We use the distillation method proposed in Hinton et al. (2015) as in Equation 1, the hard labels are found to be helpful, so nonzero $\lambda$s was suggested. In our case, we did a little tuning for $\lambda$ but didn't find nonzero $\lambda$s to be very helpful. Therefore, we report all of our results with $\lambda = 0$, i.e. only the second term involving soft labels is effective. More careful tuning of $\lambda$ should further improve the results since the searching space is strictly larger. We implemented a weighted version in our code, and we leave the choice of $\lambda$ as future work.

### A.4 The Transductive Setting and The Inductive Setting

Given $\mathcal{G}$, $\boldsymbol{X}$, and $\boldsymbol{Y}^L$, the goal of node classification can be divided into two different settings, i.e. transductive and inductive. In real applications, the former can correspond to predict missing attributes of a user based on the user profile and other existing users, and the latter can correspond to predict labels of some new nodes that are only seen during inference time. To create the inductive setting on a given dataset, we hold out some nodes along with edges connected to these nodes during training and use them for inductive evaluation only. These nodes and edges are picked from the test data. Using notation defined above, we pick the inductive nodes $\mathcal{V}_{ind}^U \subset \mathcal{V}^U$, which partitions $\mathcal{V}^U$ into the disjoint inductive subset and observed subset, i.e. $\mathcal{V}^U = \mathcal{V}_{obs}^U \sqcup \mathcal{V}_{ind}^U$. Then we can take all the edges connected to nodes in $\mathcal{V}_{ind}^U$ to further partition the whole graph, so we end up with $\mathcal{G} = \mathcal{G}_{obs} \sqcup \mathcal{G}_{ind}$, $\boldsymbol{X} = \boldsymbol{X}^L \sqcup \boldsymbol{X}_{obs}^U \sqcup \boldsymbol{X}_{ind}^U$, and $\boldsymbol{Y} = \boldsymbol{Y}^L \sqcup \boldsymbol{Y}_{obs}^U \sqcup \boldsymbol{Y}_{ind}^U$. We show the input and output of both settings using the notations below.

We visualize the difference between the inductive setting and the transductive setting in Figure 6.

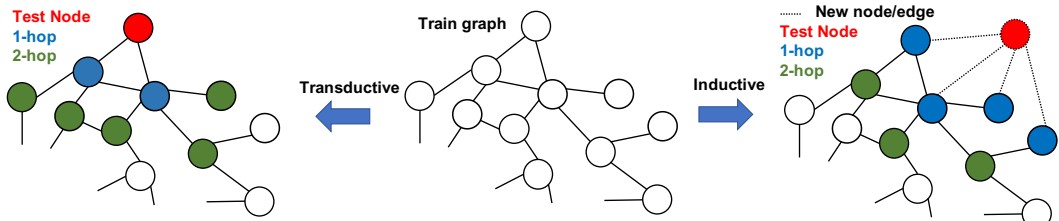

Figure 6: The transductive setting and inductive setting illustrated by a 2-layer GNN. The **middle** shows the original graph used for training. The **left** shows the transductive setting, where the test node is in red and within the graph. The **right** shows the inductive setting, where the test node is an unseen new node.

### A.5 Choosing soft targets under the transductive setting

For the transductive setting in Section 5.3, all the nodes in the graph, including the validation and test nodes, are used for the soft target generation. It seems less practical compared to the inductive case, but it is a necessary step to develop our argument. We now discuss the rationale behind this choice.

Firstly, the transductive setting is the most common setting for graph data and it was used in most GNN architecture works and GNN acceleration works we mentioned in related work. Therefore, to avoid any confusion and for a fair comparison with numbers from previous literature, we start our experiments with exactly the same input and output as the standard transductive setting. Under this setting, the inputs to GNNs include all the node features and the graph structure, so GLNN is set to be able to access the same input. As GLNN includes a teacher training step and a distillation step, the soft labels of all the nodes are intermediate outputs produced by the teacher training step, and thus used for the second distillation step for the best GLNN performance. This transductive setting can boil down to a sanity check when the student is sufficiently large. Therefore, we separate the setting to be GLNN and GLNN+ and report the results in Table 1 and Table 2 separately. In Table 1, we are checking how well GLNNs can perform compared to GNNs under the equal-parameter

constraint. The results can be interpreted as given a fixed parameter budget, whether there exists one set of parameters (one instantiation of the MLP) that can achieve competitive results as the GNN. Only when this holds, should we further investigate the more interesting and challenging inductive case as in Section 5.4.

Secondly, the task we focus on is node classification, which in many cases is considered as semi-supervised learning with very scarce labels. For example, `Pubmed` only uses 60 labeled nodes (20 per class) out of 20K nodes for training. Rather than design an advanced model that can do few-shot learning, our goal here is to leverage as much data as possible to simplify the model for more efficient inference. We thus utilize the soft pseudo-labels on all the unlabelled nodes for the best GLNN performance. In reality, when there is a large amount of separate unlabeled data, these unlabeled data can be used for GLNN distillation training and a different set of labeled data can be used for evaluation. In our case, we mimic this scenario in the inductive setting in Section 5.4.

### A.6 Implementation and Hardware Details

The experiments on both baselines and our approach are implemented using PyTorch, the DGL (Wang et al., 2019) library for GNN algorithms, and Adam (Kingma & Ba, 2015) for optimization. We run all experiments on a machine with 80 Intel(R) Xeon(R) E5-2698 v4 @ 2.20GHz CPUs, and a single NVIDIA V100 GPU with 16GB RAM.

## B Space and Time Complexity of GNNs vs. MLPs

Compared to MLP and GNN, GLNN provides a handy tool for users to trade-off between model accuracy and time complexity, which does not directly focus on space complexity. Given the space and time complexity are related, we now provide a more detailed discussion regarding these two complexities in our experiments.

In Table 1, the model comparison was between equal-sized MLPs (GLNNs) and GNNs. While fixing parameter budget to control space complexity is a standard approach when comparing models, it is not completely fair for cross-model comparison especially for MLPs vs. GNNs. To do inference with GNNs, the graph needs to be loaded in the memory either entirely or batch by batch, and may use much larger space than the model parameters. Thus, the actual space complexity of GNNs is much higher than equal-sized MLPs. From the time complexity perspective, the major inference latency of GNNs comes from the data dependency as shown in Section 4. Under the same setting as Figure 1, we show in Figure 3 **Left** that even a 5-layer MLP with 8 times wider hidden layers still runs much faster than a single-layer SAGE. Another example of cross-model comparison is Transformers vs. RNNs. Large Transformers can have more parameters than RNNs because of the attention mechanism, but they are also faster than RNNs in general, which is an important consideration in the context of inference time minimization.

In Table 1, we saw that for equal-sized comparison, GLNNs are not as accurate as GNNs on the OGB datasets. Following the discussion above and given the GLNNs used in Table 1 are relatively small (3 layers and 256 hidden dimensions) for millions of nodes in the OGB datasets, we ask whether this gap can be mitigated by increasing the MLP and thus GLNN sizes. The answer is yes as shown in Table 2.

## C Consistency Measure of Model Predictions and Graph Topology Based on Min-Cut

We introduce a metric to measure the consistency between model predictions and graph topology based on the min-cut problem in Section 5.6. The $K$-way normalized min-cut problem, or simply min-cut, partitions $N$ nodes in $\mathcal{V}$ into $K$ disjoint subsets by removing the minimum volume of edges. According to Dhillon et al. (2004), the min-cut problem can be expressed as

$$\max \frac{1}{K} \sum_{k=1}^{K} \frac{C_k^T A C_k}{C_k^T D C_k} \tag{4}$$

$$s.t. \quad C \in \{0,1\}^{N \times K}, C\mathbf{1}_K = \mathbf{1}_N$$

with $C$ being the node assignment matrix that partitions $\mathcal{V}$, i.e. $C_{i,j} = 1$ if node $i$ is assigned to class $j$. $A$ being the adjacency matrix and $D$ being the degree matrix. This quantity we try to maximize here tells us whether the assignment is consistent with the graph topology. The bigger it is, the less edges need to be removed, and the assignment is more consistent with existing connections in the graph. In Bianchi et al. (2019), the authors show that when replacing the hard assignments $C \in \{0, 1\}^{N \times K}$ with a soft classification probability $\hat{Y} \in [0, 1]^{N \times K}$, a cut loss $\mathcal{L}_{cut}$ in Equation 2 can become a good approximation of Equation 4 and be used as the measuring metric.

Table 8: GLNN predictions are much more consistent with the graph topology than MLPs. We show the $\mathcal{L}_{cut}$ values of GNNs, MLPs, and GLNN s on five CPF datasets. GLNN $\mathcal{L}_{cut}$ values become pretty close to the high $\mathcal{L}_{cut}$ values of GNNs, which were closely related to the GNN inductive bias.

| Datasets | SAGE | MLP | GLNN |
|---|---|---|---|
| Cora | 0.9347 | 0.7026 | 0.8852 |
| Citeseer | 0.9485 | 0.7693 | 0.9339 |
| Pubmed | 0.9605 | 0.9455 | 0.9701 |
| A-computer | 0.9003 | 0.6976 | 0.8638 |
| A-photo | 0.8664 | 0.7069 | 0.8398 |
| Average | 0.9221 | 0.7644 | 0.8986 |

# D EXPRESSIVENESS OF GNNS VS. MLPS IN TERMS OF EQUIVALENCE CLASSES OF ROOTED GRAPHS

In Chen et al. (2021), the expressiveness of GNNs and GA-MLPs were theoretically quantified in terms of induced equivalence classes of rooted graphs. We adopt their framework and perform a similar analysis for GNNs vs. MLPs. We first define rooted graphs.

**Definition 1** (Rooted Graph). *A rooted graph, denoted as $\mathcal{G}^{[i]}$ is a graph with one node $i$ in $\mathcal{G}^{[i]}$ designated as the root. GNNs, GA-MLPs, and MLPs can all be considered as functions on rooted graphs. The goal of a node-level task on node $i$ with label $y_i$ is to fit a function to the input-output pairs $(\mathcal{G}^{[i]}, y_i)$.*

We denote the space of rooted graphs as $\mathcal{E}$. Following Chen et al. (2021), the expressive power of a model on graph data is evaluated by its ability to approximate functions on $\mathcal{E}$. This is further characterized as the number of induced equivalence classes of rooted graphs on $\mathcal{E}$, with the equivalence relation defined as the following. Given a family of functions $\mathcal{F}$ on $\mathcal{E}$, we define an equivalence relation $\simeq_{\mathcal{E},\mathcal{F}}$ among all rooted graphs such that $\forall \mathcal{G}^{[i]}, \mathcal{G}'^{[j]} \in \mathcal{E}, \mathcal{G}^{[i]} \simeq_{\mathcal{E},\mathcal{F}} \mathcal{G}'^{[j]}$ if and only if $\forall f \in \mathcal{F}, f(\mathcal{G}^{[i]}) = f(\mathcal{G}'^{[j]})$. We now give a proposition to characterize the GNN expressive power (proof in Appendix E).

**Proposition 1.** *With $\mathcal{X}$ denotes the set of all possible node features and assuming $|\mathcal{X}| \geq 2$, with $m$ denotes the maximum node degree and assuming $m \geq 3$, the total number of equivalence classes of rooted graphs induced by an L-layer GNN is lower bounded by $\binom{|\mathcal{X}|+m-2}{m-1}^{2^L-1}$.*

As shown in Proposition 1, the expressive power of GNNs grows doubly-exponentially in the number of layers $L$, which means it grows linearly in $L$ after taking $\log(\log(\cdot))$. The expressive power GA-MLPs only grows exponentially in $L$ as shown in Chen et al. (2021). Under this framework, the expressive power of MLPs, which corresponds to a 0-layer GA-MLP, is $|\mathcal{X}|$. Since the former is much larger than the latter, the conclusion will be GNNs are much more expressive than MLPs. The gap between these two numbers indeed exists, but empirically this gap will only make a difference when $|\mathcal{X}|$ is small. As in Chen et al. (2021), both the lower bound proof and the constructed examples showing GNNs are more powerful than GA-MLPs assumed $|\mathcal{X}| = 2$. In real applications and datasets considered in this work, the node features can be high dimensional vectors like bag-of-words, which makes $|\mathcal{X}|$ enormous. Thus, this gap doesn't matter much empirically.

# E   PROOF OF THE PROPOSITION 1

To prove Proposition 1, we first define rooted aggregation trees, which is similar to but different from rooted graphs.

**Definition 2** (Rooted Aggregation Tree). *The depth-K rooted aggregation tree of a rooted graph $\mathcal{G}^{[i]}$ is a depth-K rooted tree with a (possibly many-to-one) mapping from every node in the tree to some node in $\mathcal{G}^{[i]}$, where (i) the root of the tree is mapped to node $i$, and (ii) the children of every node $j$ in the tree are mapped to the neighbors of the node in $\mathcal{G}^{[i]}$ to which $j$ is mapped.*

A rooted aggregation tree can be obtained by unrolling the neighborhood aggregation steps in the GNNs. An illustration of rooted graphs and rooted aggregation trees can be found in Chen et al. (2021) Figure 4. We denote the set of all rooted aggregation trees of depth L using $\mathcal{T}_L$. Then we use $\mathcal{T}_{L,\mathcal{X},m}$ to denote a subset of $\mathcal{T}_L$, where the node features belong to $\mathcal{X}$, and all the nodes have exactly degree $m$ ($m$ children), and at least two nodes out of these m nodes have different features. In other words, a node can't have all identical children. With rooted aggregation trees defined, we are ready to prove Proposition 1. The proof is adapted from the proof of Lemma 3 in Chen et al. (2021).

*Proof.* Since the number of equivalence classes on $\mathcal{E}$ induced by the family of all depth-L GNNs consists of all rooted graphs that share the same rooted aggregation tree of depth-L (Chen et al., 2021), the lower bound problem in Proposition 1 can be reduced to lower bound $|\mathcal{T}_L|$, which can be further reduced to lower bound the subset $|\mathcal{T}_{L,\mathcal{X},m}|$. We now show $|\mathcal{T}_{L,\mathcal{X},m}| \geq \binom{|\mathcal{X}|+m-2}{m-1}^{2^L-1}$ inductively.

When $L = 1$, the root of the tree can have $|\mathcal{X}|$ different choices. For the children nodes, we pick $m$ features from $|\mathcal{X}|$ and repetitions are allowed. This leads to $\binom{|\mathcal{X}|+m-1}{m}$ cases. Therefore, $\mathcal{T}_{L+1,\mathcal{X},m} = |\mathcal{X}|\binom{|\mathcal{X}|+m-1}{m} \geq \binom{|\mathcal{X}|+m-2}{m-1}$.

Assuming the statement holds for $L$, we show it holds for $L + 1$ by constructing trees in $\mathcal{T}_{L+1,\mathcal{X},m}$ from $T, T' \in \mathcal{T}_{L,\mathcal{X},m}$. We do this by assigning node features in $\mathcal{X}$ to the $m$ children of each leaf node in $T$ and $T'$. First note that when $T$ and $T'$ are two non-isomorphic trees, two depth-L+1 trees constructed from $T$ and $T'$ will be different no matter how the node features are assigned. Now we consider all the trees can be constructed from $T$ by assign node features of children to leaf nodes.

We first consider all paths from the root to leaves in $T$. Each path consists of a sequence of nodes where the node features form a one-to-one mapping to an L-tuple $\tau \in \{(x_1, \ldots, x_L) : x_i \in \mathcal{X}\}$. Leaf nodes are called *node under $\tau$* if the path from the root to it corresponds to $\tau$. The children of nodes under different $\tau$s are always distinguishable, and thus any assignments lead to distinct rooted aggregation trees of depth $L + 1$. The assignment of children of nodes under the same $\tau$, on the other hand, could be overcounted. Therefore, to lower bound $\mathcal{T}_{L+1,\mathcal{X},m}$, we only consider a special way of assignments to avoid over counting, which is that children of all nodes under the same $\tau$ are assigned the same set of features.

Since we assumed that at least two nodes of $T$ have different features, there are at least $2^L$ different $\tau$s corresponding to the path from the root to leaves. For a leaf node $j$ under a fixed $\tau$, one of its children needs to have the same feature as $j$'s parent node. This restriction is due to the definition of rooted aggregation trees. Therefore, we only pick features for the other $m - 1$ nodes, which will be $\binom{|\mathcal{X}|+m-2}{m-1}$ cases for each $j$. Then through this construction, the total number of depth-L+1 trees from $T$ can be lower bounded by $\binom{|\mathcal{X}|+m-2}{m-1}^{2^L}$. Finally, we have this lower bound holds for all $T \in \mathcal{T}_{L,\mathcal{X},m}$, so we derive $\mathcal{T}_{L+1,\mathcal{X},m} \geq \binom{|\mathcal{X}|+m-2}{m-1}^{2^L} \mathcal{T}_{L,\mathcal{X},m}$, and $\mathcal{T}_{L,\mathcal{X},m} \geq \binom{|\mathcal{X}|+m-2}{m-1}^{\sum_{l=1}^{L} 2^l} = \binom{|\mathcal{X}|+m-2}{m-1}^{2^L-1}$

$\square$

## F    ADVANCED GNN ARCHITECTURES AS THE TEACHER

In our experiment, SAGE teacher is used throughout to avoid influence by model architecture. Some other GNNs like GCN are also considered in the ablation studies, but they are not the best known architecture for a specific dataset. To show GLNN has stronger performance given a stronger teacher, we consider the best teacher we can access on `Products`. We take MLP+CS Huang et al. (2021) from the OGB leaderboard as a new teacher, which has reported accuracy 84.18% and ranks 8 on the leadarboard as of Nov 2021. We choose MLP+CS instead of the other top 7 because the others either rely on raw text (additional info to the given node feature), or require a large GPU with >16GB memory, which we don't have access to. Also, their improvement is not super significant compared to MLP+CS, i.e. 84% to 86%. The result with MLP+CS teacher is shown in Table 9. We see that with the new teacher, performance of GLNN+ improves to be even better than SAGE (78.61%), which shows GLNN can get stronger given a stronger teacher.

Table 9: GLNN+ with MLP+CS teacher on `Products`

|     | MLP+C&S | MLP+ | GLNN+ |
| --- | --- | --- | --- |
| Acc | 84.18 | 64.50 | 82.94 |

## G    GLNN WITH FEATURE AUGMENTATION FROM ONE-HOP NEIGHBORS

In our main experiment, the inductive performance of GLNN on the `Arxiv` dataset is less desirable than others. We thus consider augment the node features with their one-hop neighbors to include more graph information. This can be seen as a middle ground between pure GLNNs and GNNs. For this new experiment, we follow the setting in Table 3 but with two new approaches. We explain the setting of these two approaches below.

1. 1-hop GA-MLP: firstly, for each node $v$, we collect features of its 1-hop neighbors $u$ to augment the raw feature of $v$, i.e. $x_v \rightarrow \tilde{x}_v$, like in SGC. Then we train an MLP on the graph with $\tilde{x}_v$. Note if $v$ is in the observed graph but $u$ is in the inductive (unobserved during training) part, then $v$ doesn't collect features from $u$.

2. 1-hop GA-GLNN: Go through the same feature augmentation step as 1-hop GA-MLP. Then train an MLP with distillation from teacher GNN.

3. In summary, we compare 5 different models in the table below

   (a) SAGE: single model on $x_v$
   (b) MLP: single model on $x_v$
   (c) GLNN: SAGE teacher and MLP student on $x_v$
   (d) 1-hop GA-MLP: single model on $\tilde{x}_v$
   (e) 1-hop GA-GLNN: SAGE teacher on $x_v$, MLP student on $\tilde{x}_v$

We show in the table below, with 1-hop neighbor features, performance of GLNN improves a lot. This is expected as we also observe significant improvement from MLP to 1-hop GA-MLP. However, we indeed see 1-hop GA-GLNN (68.83) can further improve from 1-hop GA-MLP (66.62) and nearly match the teacher (70.64).

Table 10: GLNN with feature augmentation from one-hop neighbor on `Arxiv`

|       | Eval | SAGE | MLP | GLNN | 1-hop GA-MLP | 1-hop GA-GLNN |
| --- | --- | --- | --- | --- | --- | --- |
| `Arxiv` | *ind* | 70.64 | 55.40 | 60.48 | 66.62 | 68.83 |
|       | *tran* | 70.75 | 55.28 | 71.46 | 66.67 | 69.82 |

As we have shown in Figure 3, the 1-Layer GNN in our case is roughly 4 times slower than GLNN (29.31ms vs. 7.56ms), which should be a good approximation for the speed comparison between 1-hop GA-MLP/GA-GLNN and GLNN. This result is practically beneficial, as it gives practitioners more flexibility about how much accuracy they want to trade for less inference time.

# H   MODEL PERFORMANCE UNDER DIFFERENT INDUCTIVE SPLIT RATE

This section is a continuation of the ablation study of inductive split rate in Section 6. It generalizes Figure 5 **Middle** to more split rates (from 10:90 to 90:10), and explicitly show the inductive and transductive performance on each dataset. For better visualization, the training data label rate is also reduced from 20 per class to 5 per class in the following plots.

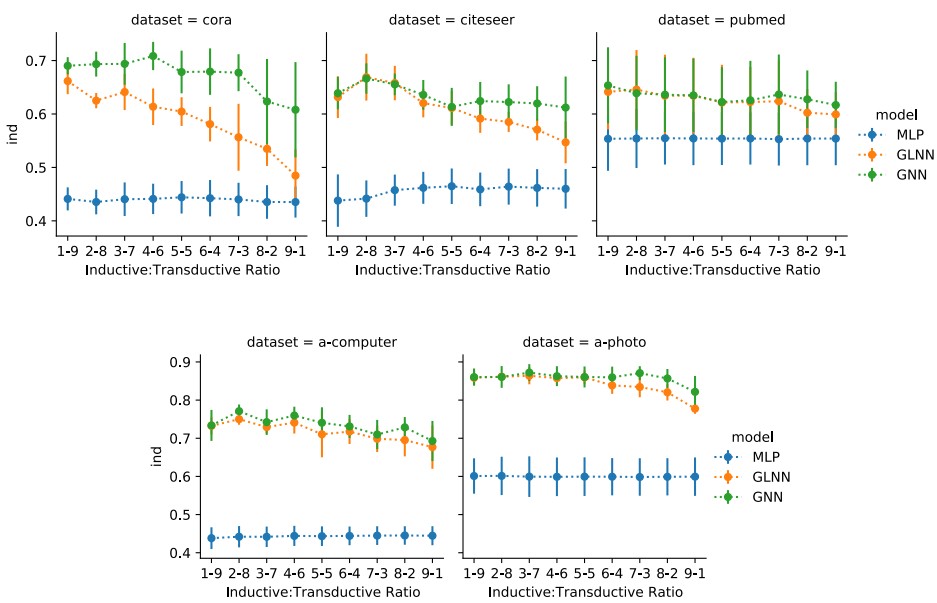

Figure 7: Model **inductive** performance comparison between MLP, GNN(SAGE), and GLNN under different inductive split rate in the production setting.

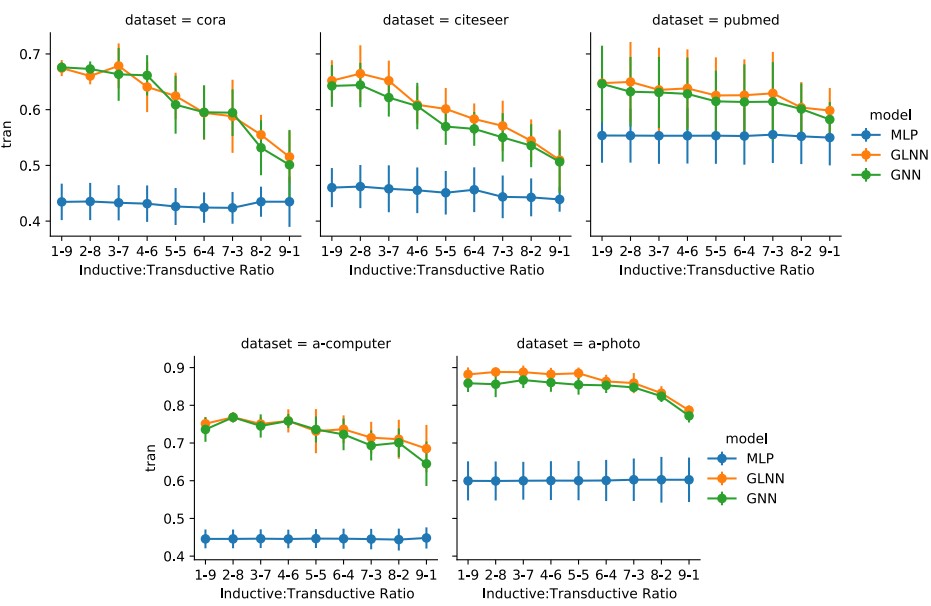

Figure 8: Model **transductive** performance comparison between MLP, GNN(SAGE), and GLNN under different inductive split rate in the production setting.

# I  GLNN UNDER NODE FEATURE HETEROGENEITY AND NON-HOMOPHILY

Besides the 7 datasets used in the main experiments, we consider 4 more datasets from Ivanov & Prokhorenkova (2021) and Lim et al. (2021) to further evaluate GLNN.

The `House_class` and `VK_class` datasets are from Ivanov & Prokhorenkova (2021). The node features of these two graphs are based on tabular data, which have different types, scales, and meanings as the opposite of the bag-of-word node features in `Cora` and etc. Some basic statistics of the datasets are shown in the following table.

Table 11: Statistics of dataset with heterogeneous node features

| Dataset | # Nodes | # Edges | # Features | # Classes |
|---|---|---|---|---|
| House_class | 20,640 | 182,146 | 6 | 5 |
| VK_class | 54,028 | 213,644 | 14 | 7 |

We apply the GLNN on `House_class` and `VK_class` using the best BGNN model from Ivanov & Prokhorenkova (2021) as the teacher. The comparison is shown in the following table. Ivanov & Prokhorenkova (2021) also includes GAT, GCN, AGNN, and APPNP as baselines, whose performance on these two datasets are quite similar (difference < 0.025). We compare with these baselines by including the best result among the 4 GNN models and refer it as GNN in the table below, i.e. GNN = max(GAT, GCN, AGNN, APPNP). From the table, we see that GLNN can improve from MLP, outperform GNN and LightGBM, and become competitive to the teacher BGNN.

Table 12: GLNN on datasets with heterogeneous node features. Numbers other than GLNN are taken from Ivanov & Prokhorenkova (2021)

| Dataset | LightGBM | GNNs | BGNN | MLP | GLNN |
|---|---|---|---|---|---|
| House_class | 0.55 | 0.625 | 0.682 | 0.534 | 0.672 |
| VK_class | 0.57 | 0.577 | 0.683 | 0.567 | 0.641 |

We further pick the non-homophilous `Penn94` and `Pokec` datasets from Lim et al. (2021). Some basic statistics of the datasets are shown in the following table.

Table 13: Statistics of non-homophilous datasets

| Dataset | # Nodes | # Edges | # Features | # Classes |
|---|---|---|---|---|
| Penn94 | 41,536 | 1,590,655 | 5 | 2 |
| Pokec | 1,632,803 | 30,622,564 | 65 | 2 |

Using the GCN teacher, we see that the performance of GLNN is improved over MLP and becomes competitive to the teacher GCN on `Penn94`. However, on `Pokec`, the simple LINK model can achieve very good performance, and it is better than most GNNs reported in Lim et al. (2021). LINK is a purely structural model which does not use node features at all. This shows that the `Pokec` dataset corresponds to the setting we discussed in Sec 5.8 (limitations of GLNN) – if the node labels can be largely determined by only the graph structure, then GLNN will struggle. We observe that GLNN is not as good as LINK owing to this limitation. However, we still see that for most of the non-homophilous datasets, MLPs already work quite well on them, and we can use GLNN for the other ones like `Penn94`.

Table 14: GLNN on non-homophilous datasets. Numbers other than GLNN are taken from Lim et al. (2021)

| Dataset | LINK | GCN | MLP | GLNN |
|---|---|---|---|---|
| Penn94 | 80.79 | 82.47 | 73.61 | 81.69 |
| Pokec | 80.54 | 75.45 | 62.37 | 61.32 |

## J    MODEL COMPARISON WITH NOISY NODE FEATURES

In Section 6, we conducted an ablation study to compare model performance with noisy node features, and the result is shown in the left plot in Figure 5. There are two subtle points in this plot. **(1)** The performance of GNN is still relatively high for high noisy features, even when $\alpha = 1$ and the features are completely random. **(2)** For completely random features, the performance of GLNN is still higher than MLP. We now discuss and explain them in more detail.

**GNN Performance on Random Features.** GNN still performs well because nodes with the same labels are likely to be connected and GNN can overfit the training data. We explain the detail through a toy example. Suppose there is a 4-clique containing nodes A, B, C, D in the graph with only a single edge D-E connects this clique to other graph nodes. Suppose A, B, C, D all have iid random Gaussian raw features and the same class label c. Let's pick A to be the inductive test node and assume E and the triangle formed by B, C, D to be in the training graph. Let's consider a simple example for 1-layer GCN and break down message passing into feature aggregation and nonlinear transformation. During training, GNN can overfit the data by learning a nonlinear transformation which maps the aggregated features of B, C, D to class c. The aggregated features of B and C will just be the average of the raw features of B, C, D. Although E is also involved in D's feature aggregation step, the aggregated features of D will also be very close to this average. Then when test on A, the aggregated feature of A will likely be classified to the same class c by the overfitted nonlinear transformation because it is the average of raw node features of A, B, C, D. In this case, GNN can actually correctly classify A because of the overfitting. For GNNs with more layers and graphs with more neighbor nodes, the conclusion may be generalized. This is roughly sort of a "majority vote" process. For a test node A, if many nodes, which A collects features from, have the same class label and appear in the training graph, then A will be classified as this class by an overfitted classifier.

**GLNN and MLP Performance on Random Features.** The gap between MLP and GLNN is due to imbalanced datasets. The GLNN can learn the imbalance from soft labels, whereas MLPs can only access uniformly picked training nodes. We explain more detail using the `A-computer` dataset as an example, for which the gap between MLP and GLNN is obvious. The task is 10-class classification. With random node features ($\alpha$=1), the inductive accuracy for MLP is 0.0652 and 0.2538 for GLNN. If the data labels are uniform, then both models should give an accuracy around 0.1. However, the labels on the inductive dataset are actually imbalanced. We show the results in Figure 9. The hist on the left is the label distribution of the inductive test set. In particular, class 4 takes about 40%. However, given this imbalance, the standard train-test split selects training nodes uniformly among labels. In this case, 20 nodes per class. Therefore, the predictions of MLP on random features are expected to be relatively uniform because the 200 nodes we train it on are uniform. This gives the hist shown in the middle, where the largest class takes about 17.5%. Finally, for GLNN, we train it on all the 200 training nodes with hard labels, plus soft labels of other nodes in the observed graph $\mathcal{G}_{obs}$ (see Section 5.2). Since these extra nodes are selected randomly, whose label distribution is actually similar to the label distribution on the whole data and the distribution on the inductive test set. Therefore, we get the GLNN predictions hist on the right. Although for each node, we can't assign a prediction correlated to its feature, on average the distribution is very close to the true label distribution on the inductive test set and has a much higher expectation. In fact, if the prediction distribution is exactly the true distribution on the inductive test set, the expectation will be 0.2169. GLNN actually does even a bit better by putting its bet more on the largest class.

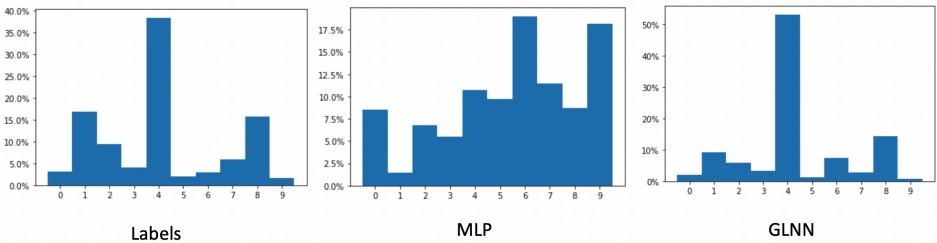

Figure 9: Inductive (predicted) label distribution on the `A-computer` dataset. **Left:** true labels. **Middle:** predicted labels by MLP. **Right:** predicted labels by GLNN.

