# OpenReview forum: "Graph-less Neural Networks: Teaching Old MLPs New Tricks Via Distillation"
_ICLR.cc/2022/Conference — ICLR 2022 Poster_

### Official Review · Reviewer_tKyT · 2021-10-20

**Correctness:** 3
**Technical Novelty And Significance:** 3
**Empirical Novelty And Significance:** 3
**Recommendation:** 8
**Confidence:** 4

**Main Review:**

The paper is overall well written and rather easy to read, with a good introduction that describes the setting of the work, several references to prior art and an ablation study that analyzes the potential failure points of the approach (which I believe provides a good understanding of the pros and cons of the solution). To the best of my knowledge, the idea presented in the manuscript is overall novel and I believe generally interesting to the community given the performance and speed up the authors managed to achieve with their approach in the analyzed settings (e.g. 273x while losing only 2% in accuracy over OGB Products).

The experimental evaluation proposed in the manuscript is generally sound, although I do have some points I would like to highlight in my review on this matter. Unless I’m misunderstanding the content of Section 5.2, the experiments for the transductive setting presented in Table 1 I believe boil down to a sanity check. As presented in the end of Section 5.2 indeed, the MLP in the transductive setting is trained to reproduce the logits of the GNN on *all* nodes of the given graph, irrespectively on whether these belong to the training, validation or test set. If the nodes thus all show different features, a sufficiently large and deep MLP should always be able to overfit on the features of the target nodes and replicate the predictions of the GNN. As such, the experiments presented in Table 1 don’t look particularly meaningful to describe the quality of the proposed approach and I would ask the author to please clarify in their rebuttal whether they avoided training the MLP on the logits produced by the GNN for validation / test nodes and, if I’m right, to better clarify the relevance of these results in the text.

Following the previous comment, in Table 4 the authors present two aggregated measures of the quality of the model (\Delta_{MLP} and \Delta_{GNN}) that should describe a typical production setting consisting of 80% unlabeled nodes that have been observed at training time and 20% of new unlabeled nodes the model has never seen before. As for the above, I believe the most interesting result here is represented by the performance of the model in the inductive scenario and I would thus suggest the authors to express the improvements in performance in the table for all the 3 different settings they described (i.e. transductive, inductive and production) instead of just the production one (it would better highlight the performance of the approach in different conditions and thus enhance clarity).

Finally, Figure 5 middle, it is possible to observe how the GNN actually improves in performance over the inductive set if we increase the size of this and thus restrict the size of the transductive one (i.e. the number of nodes the model is actually able to observe at training time). While the improvement seems to be marginal from the picture, this is somehow a surprising and counterintuitive result and I wonder if the authors have an explanation for this. Additionally, I would recommend adding standard deviation bars in all the plots of Figure 5 to highlight the stability of the performance of the models over the different datasets and potentially have a different picture (or a series of pictures) in the supplementary material showing the curves for each dataset independently.

Besides the points highlighted above, I believe the manuscript is in good shape and I would recommend the acceptance of the paper to the conference.


**Summary Of The Paper:**

The paper proposes a Knowledge Distillation (KD) based approach for producing MLPs able to achieve comparable performance to Graph Neural Networks (GNNs) on node-wise classification tasks. The MLPs receive as input only the features of each target node v and it is trained to imitate the behavior of a pre-trained GNN on an extended version of the training set consisting of:

1) the original labeled set where the GNN was trained;
2) an additional set of unlabeled nodes which are equipped with soft labels produced by the GNN.

As the GNN (which operates as a teacher in this scenario) has a good inductive bias, it can learn a robust decision boundary even from a rather small training set. Such a decision boundary can then be taught (at least partially) to the simpler MLP (which operates as a student) by training the MLP on a larger training set where it learns to capture patterns in the target nodes features that correlate well with the soft labels produced by the GNN. If the GNN shows good performance over unseen condition, the MLP trained in this way ends up being more robust wrt a MLP trained only on the original labeled set as it can observe a potentially significantly wider variety of conditions at training time.

As in this scenario the MLP doesn’t receive any descriptor representing the behavior of the neighborhood of the target nodes, the overall approach doesn’t work in all possible settings (e.g. in situations where the label of a node depends exclusively on the behavior of the neighbors). Nonetheless, the authors showed in the paper that in classic scenarios (i.e. node classification tasks on citation or product networks) it is possible to effectively train a MLP able to achieve comparable performance to a GNN, while requiring a fraction of the computational cost as no convolution is realized on the given graph.

**Summary Of The Review:**

The paper proposes a novel approach for effectively training MLPs for node-wise classification tasks on graph structured data. The proposed approach allows to train architectures able to approach in most scenarios the performance of a reference GNN (SAGE) while requiring a fraction of the computational cost as no convolution is realized on the given graph. The experimental evaluation presented in the paper is generally sound and well highlights the benefits of the idea proposed in the manuscript.

---

> ### Author Response · Authors · 2021-11-23
> **Response**
>
> Dear Reviewer tKyT,
>
> We really appreciate your comments and your support for our work. We hope our response can address your concerns. Please find our detailed response below.
>
> 1. Experiment results in Table 1 boil down to a sanity check, and the most interesting result comes from the inductive scenario.
>
> Yes, your understanding of Table 1 is correct. As shown in the formula at the end of Sec 5.2, we leverage the logits of all the nodes in the graph for the distillation. There are two reasons to adopt this setting.
>
> * Firstly, the transductive setting is the most common setting for graph data and was used in most GNN architecture and GNN acceleration works, e.g. the CPF [1], where we take the dataset and model hyperparameters from. Therefore, to avoid any confusion and for a fair comparison with numbers from previous literature, we start our experiments with the exact same input/output as the standard transductive setting. Under this setting, the input to GNN includes all the node features and the graph structure, so GLNNs are set to be able to access the same input. GLNN includes a teacher training step and a distillation step. The soft labels of all the nodes are intermediate outputs produced by the teacher, and thus used for the second distillation step for the best GLNN performance. As you mentioned, this setting boils down to a sanity check when the MLP is sufficiently large. For this reason, we divide the setting to be GLNN and GLNN+ with results in Table 1 and Table 2 separately. In Table 1, we check how well GLNNs can do compared to GNNs under the equal-parameter constraint. It can be interpreted as given a fixed parameter budget, whether there exists a set of parameters (one instantiation of the MLP) that can achieve competitive results as the GNN. Only when this holds, should we further investigate the more interesting and challenging inductive case.
>
> * Secondly, we focus on node classification, which is often considered as semi-supervised with scarce labels. For example, Pubmed only uses 60 labeled nodes (20 per class) out of ~20K nodes. Rather than design an advanced model for few-shot learning, our goal is to leverage as much data as possible to simplify the model for more efficient inference. We thus utilize the soft pseudo-labels on all the unlabelled nodes for the best GLNN performance. In reality, when there is a separate set of unlabeled data, they can be used for GLNN distillation and a different set of labeled data can be used for evaluation. In our case, we mimic this scenario in the inductive setting.
>
> We totally agree that the inductive setting is the most interesting one. As shown in Figure 2, it is the major use case of GLNN we aimed for. We propose the production setting because it is closer to the real-world scenario: Good performance should be maintained for both the observed nodes in the graph and new inductive nodes. This is also why we only had delta values for the production setting - to make the table easier to read since delta columns are not super common in most paper tables. Nevertheless, we agree with your suggestion to further clarify these settings, so we have done the following.
>
> * Add clarification in the text of what nodes are used for soft target generation under different settings in Sec 5.2. Rephrase this rebuttal response and add it in the appendix for the reader’s reference.
>
> * Add the full version of Table 3 with the explicit Delta values for all three settings.
>
> Thank you for making these suggestions to help us increase paper clarity.
>
> 2. Figure 5 middle, inductive performance increases while we increase the inductive size.
>
> The marginal improvement in inductive performance as the inductive set size increases is due to randomness. As the inductive rate increases, model performance roughly stays the same when the inductive rate is between 10% - 50% and the label rate is relatively high, i.e. 20 labeled nodes per class for training and 30 labeled nodes per class for validation. We show new experimental results to clear this confusion.
>
> We create more plots in Appendix H, where training labels are reduced to 5 per class, and results are shown from ind:tran = 10:90 to ind:tran = 90:10. We follow your advice to make a series of plots for each dataset and add standard deviation bars, which is a great way to show that some marginal improvements are due to randomness. We don’t plan to add these bars to Figure 5 since it will be too crowded. Rather we add a sentence to direct readers to the Appendix. From plots in Appendix H, we see the trend of performance decreasing for both transductive and inductive settings on all datasets. The decrease is not too obvious when the inductive rate is low. Also, the transductive performance decreases less than inductive because inductive evaluation is on the full graph and transductive evaluation is on the observed graph.
>
> Please let us know if there are any remaining concerns not addressed. We appreciate any feedback.

---

> > ### Comment · Reviewer_tKyT · 2021-11-25
> > **Response**
> >
> > Dear authors,
> > thank you for your comments, they clarify my main doubts. I'm pleased to inform you that I thus confirm my score and I recommend the acceptance of the paper to the conference. Well done!

---

> > > ### Author Response · Authors · 2021-11-29
> > > **Thank you**
> > >
> > > We truly appreciate your effort in helping us to strengthen the paper and your support for our work!

---

### Official Review · Reviewer_fkQB · 2021-10-29

**Correctness:** 4
**Technical Novelty And Significance:** 4
**Empirical Novelty And Significance:** 4
**Recommendation:** 10
**Confidence:** 4

**Main Review:**

The paper shows that it's possible to distill knowledge from GNNs to MLPs without sacrificing performance, while reducing inference time by orders of magnitude. The approach is very simple, which copies a classical KD approach proposed by Hinton et al. At inference, it shows significant reduction of time compared to baseline GNNs and existing approaches. The paper is well-written, motivated, without unnecessary theory, and does enough ablation studies.

The only concern is the datasets used to verify the approaches. While there are OGB ones, cora/citeseer/pubmed are quite small for this type of approaches to motivate enough KD. I would include other datasets, for example, coming with heterogeneous node features [1] (where MLP and GBM can work quite well) and heterophilous datasets [2] (where LP can work well). Results there would give additional insights onto the use cases of KD for GNNs. It's also interesting to compare performance of this approach compared to graph-agnostic baseline using GBM [1] (it should give even less inference time than MLPs and as such could be more amenable for this type of KD).

I may update my score based on the authors' responses.

[1] Boost then Convolve: Gradient Boosting Meets Graph Neural Networks https://arxiv.org/abs/2101.08543
[2]  New Benchmarks for Learning on Non-Homophilous Graphs  https://arxiv.org/abs/2104.01404

**Summary Of The Paper:**

The paper proposes to distill knowledge from GNNs to MLPs so that at inference time it requires less burdensome. The idea is neat and the implementation is flawless.

**Summary Of The Review:**

Practical limitations of GNNs in industry setup are quite severe for existing GNN models and this approach offers an easy way to overcome them and leverage graph topology at the same time.

---

> ### Author Response · Authors · 2021-11-23
> **Response**
>
> Dear Reviewer fkQB,
>
> We really appreciate your comments and your support for our work. We hope our response can address your concerns. Please find our detailed response below.
>
> As you suggested, we also ran our GLNN framework to show new results on datasets with heterogeneous node features[1] and non-homophilous datasets [2]. We take these datasets from [1] and [2] and adopt the exact same settings as the corresponding paper for each dataset (data split, evaluation metric, and etc.), so the numbers can be directly compared to the original paper.
>
> * In [1], other than the OGB-ArXiv we have already included, there are four datasets with heterogeneous node features for the classification task. According to the discussion in the second paragraph on top of page 7 in [1], graph structure was found to be not very helpful for the Slap and DBLP datasets. Therefore, inference efficiency is not an urgent problem as non-graph-dependent models can be used for these two datasets, e.g. GBM shows the best performance among all models in [1]. We thus consider the remaining House_class and VK_class datasets, where we use the best BGNN model from the paper as the teacher and MLP as the student. The comparison is shown in the following table. The paper also includes GAT, GCN, AGNN, and APPNP as baselines, whose performance on these two datasets is quite similar (difference < 0.025). We compare with these baselines by including the best result among the 4 GNN models and refer it as GNN in the table below, i.e. GNN = max(GAT, GCN, AGNN, APPNP).  From the table below, we see that GLNN can improve from MLP, outperform GNN and LightGBM, and become competitive with the teacher BGNN.
>
> |Dataset|LightGBM|GNNs|BGNN|MLP|GLNN|
> |-|-|-|-|-|-|
> |House_class|0.55|0.625|0.682|0.534|0.672|
> |VK_class|0.57|0.577|0.683|0.567|0.641|
>
> * In [2], there are eight homophily datasets. For many of them, MLP can already outperform or achieve competitive performance as GNNs. We thus pick two datasets Penn94 and pokec, where there is a ~10% gap between the performance of MLP and GNN. Using the GCN teacher, we see that on the Penn94 datasets, the performance of GLNN is improved over MLP and becomes competitive to the teacher GCN. However, on the pokec dataset, the simple LINK model can achieve very good performance, and it is better than all GNNs reported in the paper. The LINK model is a purely structural model which does not use node features at all. This shows that the pokec dataset corresponds to the setting we discussed in Sec 5.8 (limitations of GLNN) -- if the node labels can be largely determined by only the graph structure, then GLNN will struggle. We observe that GLNN is not as good as the LINK model owing to this limitation.  However, we still see that for most of the non-homophily datasets, MLPs already work quite well on them, and we can use GLNN for the other ones like Penn94.
>
> |Dataset|LINK|GCN|MLP|GLNN|
> |-|-|-|-|-|
> |Penn94|80.79|82.47|73.61|81.69|
> |pokec|80.54|75.45|62.37|61.32|
>
> We have added these tables and corresponding discussions in our Appendix I. Thank you for helping us strengthen the experiments.
>
> Please let us know if there are any remaining concerns not addressed. We appreciate any feedback.
>
> **Reference**
>
> [1] I​​vanov, S., & Prokhorenkova, L. (2021). Boost then Convolve: Gradient Boosting Meets Graph Neural Networks.
>
> [2] Lim, D., Li, X., Hohne, F., & Lim, S. N. (2021). New Benchmarks for Learning on Non-Homophilous Graphs.

---

> > ### Comment · Reviewer_fkQB · 2021-11-29
> > **Good job.**
> >
> > I thank the authors for the responses. I'm satisfied with the paper, I like both the idea and the implementation. The authors have addressed my concerns and I, therefore, increase my score and recommend acceptance.

---

> > > ### Author Response · Authors · 2021-11-29
> > > **Thank you**
> > >
> > > We truly appreciate your effort in helping us to strengthen the paper and your support for our work! We are so humbled to receive such recognition.

---

### Official Review · Reviewer_UK5z · 2021-11-02

**Correctness:** 4
**Technical Novelty And Significance:** 3
**Empirical Novelty And Significance:** 3
**Recommendation:** 8
**Confidence:** 3

**Main Review:**

* Pros:

1. The motivation of training student MLPs with teacher GNNs is very important in practice. This paper explores a novel path that is separate from typical acceleration techinique in GNNs, such as sampling and canceling nonlinear transformation. Related discussion in Section 5.5 is well made.
2. The careful discussion on transductive and inductive learning is significant, with comprehensive experiment results in Table 3. I thought it would not work well in inductive cases, but it can.
3. It is an intriguing discovery that GLNNs with larger widths can achieve considerable performance boost.
4. The teacher GNN architecture is flexible, as shown in Section 6.

* Concerns:

1. To clarify, is the training loss of GLNNs in Figure 4 the same as the first label-loss term in (1)?
2. There is a related work [1] that also trains MLPs on graphs instead of GNNs. They train MLPs with contrastive loss based on graph structure to guide MLPs to find good parameters in the MLP space, which is sharing similar spirits. It would be good to have a discussion on that.
3. I am wondering whether it will enhance the performance further if the input feature is combined with 1-hop neighorhood, such summation of features in 1-hop neighneighorhood, which is very simple. (Or compare with pure 1-hop GA-MLPs.) This stands between GLNNs and GA-MLP models. The motivation is from the fact that usually low-order neighborhood is the best choice and GNNs cannot handle well with long-range information. In this sence, it will be interesting to see how GLNNs work with simply combined features from low-order neighborhood, under the supervision of GNNs.

4. Since MLPs are trained to fit GNNs in different architectures, is it possible to explore the bias of different GNNs via analyzing learned parameters/functions in MLPs?

[1] Graph-MLP: Node Classification without Message Passing in Graph. Yang Hu et al. ArXiv:2106.04051.

**Summary Of The Paper:**

To overcome inference latency of GNN models, this paper proposes a GLNN framework that trains a student MLP with supervision from a teacher GNN model. GLNNs significantly improves performances of pure MLPs and are comparable with traditional GNNs in many cases. The paper explores different real-world settings including transductive and inductive learning. It also discusses when GLNNs will fail, which is a significant question, and address that it is rare in practice.

**Summary Of The Review:**

I would like to recommend to accept this paper for its practical intuition, satisfying performance and comprehensive discussion on limitations.

---

> ### Author Response · Authors · 2021-11-23
> **Response (2/2)**
>
> 4. Is it possible to explore the bias of different GNNs via analyzing learned parameters/functions in MLPs?
>
> This is indeed an interesting question that requires careful investigation. Firstly, it is not immediately obvious that comparing the learned GLNN (or MLP)’s parameters is a better option to compare multiple teacher GNNs’ biases, rather than e.g. directly adopting a representational similarity lens as in [2], which is applicable across networks. Moreover, if adopting the suggested approach, we may have to account for differences in the teacher GNN’s expressivity, convergence, and their own parametrization which will likely impact the quality of the student in a way that might be non-trivial to disentangle, making it hard to attribute differences to the teacher’s bias, rather than these other properties. Nevertheless, we appreciate the insightful question and look forward to giving it some more careful thought.
>
> Please let us know if there are any remaining concerns not addressed. We appreciate any feedback.
>
> **Reference**
>
> [1] Graph-MLP: Node Classification without Message Passing in Graph. Yang Hu et al. arXiv:2106.04051 (2021)
>
> [2] Insights on Representational Similarity in Neural Networks with Canonical Correlation. Morcos et al. NeurIPS’18.

---

> > ### Comment · Reviewer_UK5z · 2021-11-29
> > **Reviewer response**
> >
> > My concerns are addressed. But it would be better if the authors could reveal the performance with different hidden dimensions of teacher networks.
> >
> > For now I would like to hold this score.

---

> > > ### Author Response · Authors · 2021-11-29
> > > **Thank you**
> > >
> > > We truly appreciate your effort in helping us to strengthen the paper and your support for our work!
> > >
> > > For the teacher hyperparameters, we want to avoid their influence and make our numbers directly comparable to previous works. Therefore, we follow the hyperparameters in published implementations, which were carefully tuned either manually or by automatic hyperparameter search frameworks, e.g. Optuna. Nevertheless, more teachers with different hidden dimensions will for sure strengthen the paper. We will try to include more of these experiments in our next version.

---

> ### Author Response · Authors · 2021-11-23
> **Response (1/2)**
>
> Dear Reviewer UK5z,
>
> We really appreciate your comments and your support for our work. We hope our response can address your concerns. Please find our detailed response below.
>
> 1. Whether training loss in Figure 4 is the same as the label loss in Equation (1)
>
> Yes. These two losses are the same as you pointed out since the loss of the soft targets is not directly comparable to the validation loss or loss for MLPs. We have highlighted in red a clarification in the caption of Figure 4.
>
> 2. Related work Graph-MLP
>
> Thank you for pointing us to this related work. We have added Graph-MLP in our related work section to discuss the method and its difference from ours. We highlight the text in red in the updated PDF and also give it below.
>
> “ Graph-MLP also tries to bypass GNN neighbor fetching. It trains an MLP with an additional neighbor contrastive loss (Ncontrast) to implicitly incorporate graph structure information. Ncontrast pushes together embeddings for neighboring nodes and pulls apart embeddings for far away nodes. Our approach is different from Graph-MLP as we utilize distillation to incorporate graph structure information from a teacher GNN.”
>
> According to Table 2 in the Graph-MLP paper, its performance is competitive to many GNNs (GCN, GAT, and etc) on Cora, Citeseer, and Pubmed under the transductive setting. For GLNN, we show that it can achieve competitive results on these datasets and also OGB datasets under both transductive and inductive settings. Another advantage of GLNN over the Graph-MLP is that GLNN can get stronger given a stronger teacher as we show in Appendix F. We further show in Appendix I that GLNN can work well on graphs with heterogeneous node features and non-homophilous graphs when proper models are used as the teacher. While more advanced GNNs are being developed, GLNN can enjoy the benefit by distilling from these new models. In contrast, the assumption of Graph-MLP, i.e. nodes within r-hops should be similar to each other, may not hold for non-homophilous graphs.
>
>
> 3. Performance enhancement if the input features are augmented with 1-hop neighbors (1-hop GA-MLPs).
>
> Thank you for suggesting this interesting experiment. Intuitively, aggregating 1-hop neighbor features should improve the performance. To verify it, we experiment on the ArXiv dataset under the production setting (setting of Table 3). We omit other settings and datasets for now since GLNN has surprisingly good results for those cases and can often match the teacher’s performance, thus the improvement won’t be very obvious. We describe the two new methods in this experiment in detail.
>
> * 1-hop GA-MLP: firstly, for each node $v$, we collect features of its 1-hop neighbors $u$ to augment the raw feature of $v$, i.e. $x_v \rightarrow \tilde x_v$, like in SGC. Then we train an MLP on the graph with  $\tilde x_v$. Note if $v$ is in the observed graph but $u$ is in the inductive (unobserved during training) part, then $v$ doesn’t collect features from $u$.
>
> * 1-hop GA-GLNN: Go through the same feature augmentation step as 1-hop GA-MLP. Then train an MLP with distillation from teacher GNN.
>
> * In summary, we compare 5 different models in the table below
>     * SAGE: single model on $x_v$
>     * MLP: single model on $x_v$
>     * GLNN: SAGE teacher and MLP student on $x_v$
>     * 1-hop GA-MLP: single model on $\tilde x_v$
>     * 1-hop GA-GLNN: SAGE teacher on $x_v$, MLP student on $\tilde x_v$
>
> The inductive performance of GLNN on the ArXiv dataset was not as good as the teacher. We show in the table below, with 1-hop neighbor features, the performance of GLNN improves a lot. This is expected as we also observe significant improvement from MLP to 1-hop GA-MLP, but we indeed see 1-hop GA-GLNN (68.83) can further improve from 1-hop GA-MLP (66.62) and nearly match the teacher (70.64).
>
> |Dataset|Eval|SAGE|MLP|GLNN|1-hop GA-MLP|1-hop GA-GLNN|
> |-|-|-|-|-|-|-|
> |ArXiv|ind|70.64|55.40|60.48|66.62|68.83|
> ||tran|70.75|55.28|71.46|66.67|69.82|
>
> As we have shown in Figure 3, the 1-Layer GNN in our case is roughly 4 times slower than GLNN (29.31 vs. 7.56), which should be a good approximation for the speed comparison between 1-hop GA-MLP/GA-GLNN and GLNN. This result is practically beneficial, as it gives practitioners more flexibility about how much accuracy they want to trade for less inference time. We have also included the table above in the Appendix. Thank you for helping us strengthen the experiments.

---

### Official Review · Reviewer_5DSq · 2021-11-07

**Correctness:** 3
**Technical Novelty And Significance:** 1
**Empirical Novelty And Significance:** 1
**Recommendation:** 3
**Confidence:** 5

**Details Of Ethics Concerns:**

I do not believe this work has a negative social impact or any ethics concerns.

**Main Review:**

# Interesting point
- The MLP with a simple knowledge distillation design applied on top of a teacher GNN does show huge potentials in terms of the inference time. The aspect of bypassing fully the neighborhood fetch process during inference time is interesting.

- Empirically, they show this simple design has competitive performance as the teacher GraphSAGE model while enjoying hundreds of times faster inference time and ~30x faster than other inference acceleration methods. It does show the potential of this direction.


# Main concern

**Methodology aspect**

-  The novelty of the work is too limited, they only apply the original distillation formulation from Hinton et al. 2015.  There is no interesting distillation design that tailors to the unique characteristics of the dependency complication introduced by the graph.

- The literature discussion is not thorough, missing out on quite some important related works which discuss the knowledge distillation in the context of graph-structured data [1] [2] [3].

[1] Yang, Y., Qiu, J., Song, M., Tao, D. and Wang, X., 2020. Distilling knowledge from graph convolutional networks. In Proceedings of CVPR.

[2] Deng, X. and Zhang, Z., 2021. Graph-Free Knowledge Distillation for Graph Neural Networks. arXiv preprint arXiv:2105.07519

[3] Xu, Y., Zhang, Y., Guo, W., Guo, H., Tang, R. and Coates, M., 2020, October. Graphsail: Graph structure aware incremental learning for recommender systems. In Proceedings of CIKM.

- Some claim discussed in the motivation section is not accurate. The authors claim that "To infer for a single node with a L-layer GNN on a graph with average degree R will require $O(R^L)$ fetchings", which is not accurate considering when working on large graphs, we will not use node wise sampling technique. People tend to use efficient batching and layerwise or graph-wise sampling techniques [4] [5] to greatly reduce the node fetching process.

[1] Chiang, W.L., Liu, X., Si, S., Li, Y., Bengio, S. and Hsieh, C.J., 2019. Cluster-GCN: An Efficient Algorithm for Training Deep and Large Graph Convolutional Networks. In Proceedings of KDD.

[2] Zeng, H., Zhou, H., Srivastava, A., Kannan, R. and Prasanna, V., 2020. GraphSAINT: Graph Sampling-Based Inductive Learning Method. In Proceedings of ICLR.

- Do we really have a scenario that we can establish the existing efficient batching and graph-wise sampling techniques are unable to meet the inference time constraint? Can authors provide an intuitive example in practice?

**Experimental result**

- The proposed approach doesn't show competitive performance on  Arxiv and Product dataset, which are the larger datasets we care more about the inference efficiency potentially.

- This simple distillation design didn't show competitive performance when the teacher model is more advanced GNN architecture, as shown in Figure 5 (c), which makes the technique less interesting. I believe establishing that distilling a more competitive teacher GNN into a student model with less performance drop will make the work much stronger.

- To add on top of my argument above, the top model performance (which also uses a fashion of KD) on Arvix and Product is 76.2% and 86.4% respectively. Compared to the GLNN performance on Arvix and Product which is 63.5% and 68.9%. I'm not too sure about the accuracy and computation complexity trade-off make sense.

- The baseline comparison of the paper is not thorough as I point out above regarding missing out on quite some important related works about KD in the context of graphs.  For an empirical-based paper, I do expect the experiment section to be much more thorough than what is being provided.


**Summary Of The Paper:**

This paper applies KD in the context of the graph. It aims to distill the teacher output of a GNN model into a simple MLP model. Empirically, they show that this simple KD design is able to improve the student MLP model by a large margin and can match the results coming from a teacher GNN model. Besides, it shows empathically, the inference time can be greatly improved.

**Summary Of The Review:**

The novelty of the work is limited, there is no interesting distillation design that tailors to the unique characteristics of the dependency complication introduced by the graph. Besides, the literature discussion is not thorough, missing out on quite some important related works (KD on graph-structured data). In the experimental study, this simple distillation design didn't show competitive performance when the teacher model is more advanced GNN architecture. Overall, I recommend rejecting this paper.

---

> ### Author Response · Authors · 2021-11-17
> **Response (3/3)**
>
> 6. GLNN performance doesn’t improve on more advanced GNNs as in Figure 5 (c)
>
> * For Figure 5 (c), we considered the performance on five datasets from CPF [7]. We didn’t include the OGB datasets as some GNNs won’t scale. For all 4 GNN teachers, i.e. GCN, GAT, SAGE, and APPNP, we follow the hyperparameters used in CPF, which were searched on a large grid via the automated hyperparameter search framework Optuna. It turns out that the performance of these 4 GNNs, with the best hyperparameters, are not too different when averaged over these five datasets. Therefore, it is not the case that GLNN can’t match more advanced GNNs, but the advanced GNNs don’t show advantages on these five datasets. Actually, we see from 5 (c) that GLNN can nearly match each teacher (< 1.5% drop). In the question below, we show that GLNN can match top GNN teachers on the larger OGB Products.
>
> 7. GLNN doesn’t match the top models on the OGB leaderboard.
>
> * For arXiv, as we have explained in response #5, GLNN+ can achieve 72.15%, which is even higher than the teacher SAGE (70.92%) and close to the top model without raw text on the OGB leaderboard (74.31%).
>
>
> * For Products, we report new results. In the paper, SAGE teacher is used for all experiments to avoid influence by model architecture. To show GLNN has stronger performance given a stronger teacher, we take MLP+C&S from the OGB leaderboard as a new teacher, which ranks #8 currently (84.18%). We choose MLP+C&S instead of the other top 7 because the others either rely on raw text (additional info to the given node feature), or require a GPU with >16GB memory, which we don’t have access to. Also, their improvement is not super significant compared to MLP+C&S, i.e. ~84% to ~86%. The result is below:
> |MLP+C&S| MLP+| GLNN+|
> |--|--|--|
> |84.18|64.50|82.94|
>
>
> * With the new teacher, GLNN+ is even better than SAGE (78.61%), which shows GLNN can get stronger given a stronger teacher. We have included this table in our Appendix F. Thank you for helping us to make the paper more complete.
>
> 8. Missing comparison with graph knowledge distillation (KD) baselines
>
> * As mentioned in response #2, we have added a detailed discussion regarding the KD baselines in the related work. However, as we have discussed in Sec 5.5 when we compare GLNN with different model inference acceleration methods, the comparison with other GNN-to-GNN KD baselines is not an absolute necessity for making our main point for the reasons below.
>
>
> * For accuracy comparison, the accuracy of the GNN-to-GNN KD baselines is upper bounded by the teacher GNN [3,6]. We also see that GLNN can match the teacher’s performance in most cases. Therefore, these baselines don’t show advantages over GLNN in terms of accuracy.
>
>
> * For inference time comparison, the inference speed of the distilled L-layer student in any of the GNN-to-GNN KD baselines is slower than an L-layer GNN because there will usually be overheads introduced by some extra modules like the LSP [3] or PAM [6]. Thus, comparing GLNN+ with an L-layer GNN will suffice, which we have already shown in Figure 3 that GLNN is much faster (7.56 ms vs. 29.31 ms).
>
> In light of the above, we hope you reassess our manuscript -- please let us know if there are any remaining concerns not addressed. We appreciate any feedback.
>
> **Reference**
>
> [1] Hinton, G., Vinyals, O., & Dean, J. Distilling the knowledge in a neural network
>
> [2] C. Bucila, R. Caruana, and A. Niculescu-Mizil. Model compression
>
> [3] Yang, Y., Qiu, J., Song, M., Tao, D. and Wang, X. Distilling knowledge from graph convolutional networks
>
> [4] Deng, X. and Zhang, Z. Graph-Free Knowledge Distillation for Graph Neural Networks
>
> [5] Xu, Y., Zhang, Y., Guo, W., Guo, H., Tang, R. and Coates, M. Graphsail: Graph structure aware incremental learning for recommender systems
>
> [6] Yan, B., Wang, C., Guo, G., and Lou, Y., Tinygnn: Learning efficient graph neural networks
>
> [7] Yang, C., Liu, J., and Shi, C.. Extract the knowledge of graph neural networks and go beyond it
>
> [8] Ying, R., He, R., Chen, K., Eksombatchai, P., Hamilton, W. L., & Leskovec, J. Graph convolutional neural networks for web-scale recommender systems
>
> [9]  https://eng.uber.com/uber-eats-graph-learning/
>
> [10] Zhang, D., Huang, X., Liu, Z., Hu, Z., Song, X., Ge, Z., ... & Qi, Y. Agl: a scalable system for industrial-purpose graph machine learning
>
> [11] Aravind Sankar, Yozen Liu, Jun Yu, and Neil Shah. Graph Neural Networks for Friend Ranking in Large-scale Social Platforms
>
> [12] https://www.gigaspaces.com/blog/amazon-found-every-100ms-of-latency-cost-them-1-in-sales
>
> [13] https://www.prnewswire.com/news-releases/akamai-online-retail-performance-report-milliseconds-are-critical-300441498.html
>
> [14] Dang V., Bendersky M., Croft W.B. (2013) Two-Stage Learning to Rank for Information Retrieval

---

> > ### Comment · Reviewer_5DSq · 2021-11-30
> > **Thank you**
> >
> > Dear authors, thanks for the detailed reply and apologize for my tardy response. I have carefully read your responses as well as the discussion with other reviewers.
> >
> > **Limited novelty aspect**
> >
> > - My concern about the limited novelty of this work still hold.
> >
> > - The author claims that using a stronger GNN teacher model’s prediction label to guide the learning of a weak student model shows improvement over the vanilla MLP student model is a surprising discovery. However, in my opinion, it is not at all new or surprising, since the same conclusion has been made in various applications (model compression, incremental learning, etc.) and numerous papers that the student classifiers can learn much faster and more reliably if trained with the outputs of another stronger classifier as soft labels. There is even theoretical work that brings a theoretical explanation of this phenomenon [1]. I do not believe this paper brings anything new to the table.
> >
> > - Even the case study provided in section 5.6 does not provide anything new in the sense that you distill the prediction label of a graph topology sensitive model, you will expect that after distillation the pseudo label from the teacher, the student predictions are much more consistent with graph topology than predictions of a standalone MLP. There is exactly what we should expect given Hilton’s knowledge distillation framework.
> >
> > - I do appreciate that in the modification on the related work section to properly highlight other prior works regarding the usage of the knowledge distillation in the context of GNN models
> >
> > [1] Phuong, M. and Lampert, C., 2019, May. Towards understanding knowledge distillation. In International Conference on Machine Learning (pp. 5142-5151). PMLR.
> >
> > **Comparison to advanced GNNs**
> >
> > - I appreciate the new results with MLP+C&S as the teacher model is being presented. However, one thing that is a bit hard to justify about this work is that MLP+C&S can already achieve 100 times faster for training and inference time, and has 137 times fewer parameters [2]. Compared to this alternative MLP +label propagation framework, the proposed GLNN+ has around 2% performance drop on the Arvix dataset. Is it even necessary to take the prediction of the teacher MLP+C&S and distill its prediction to an MLP student to further achieve a bit of inference time gain with the further loss of accuracy?  while the original teacher MLP+C&S already has huge advantages in both training, inference time as well as model performance.  Besides, I think this work should be highlighted more in the paper since it aims to solve a similar time computation efficiency issue (for both the training and inference phase).
> >
> > [2] Huang, Q., He, H., Singh, A., Lim, S.N. and Benson, A.R., 2020. Combining label propagation and simple models out-performs graph neural networks. ICLR 21.
> >
> >
> > **Other issues**
> >
> > - I do not think the authors have provided a justified claim regarding alternative scalable GNN designs. For example, the claim that “that node-wise sampling is still a common/de-facto approach in many practical scenarios”, firstly, this is not necessarily true in large scale landing projects in industrial setups. Secondly, even if we need to use node-wise sampling in practice, a proper feature fetching mechanism (caching the sampled neighbor information for each node) will be incorporated to reduce the training and inference time.
> >
> > - The author also claims that the work has a great practical impact and a huge trade-off is beneficial in practice. However, I do not think this work provides enough evidence to support the claim. I do think the 100x improvement in terms of the inference time aspect could be interesting. However, to the best of my knowledge, the scenarios that will require to use of GNN on a resource constraint device/case are quite limited. The urging requirement for speedup GNN is on the training phase, not the inference phase, such as in a large-scale recommendation system.
> >
> >
> >
> > Overall, considering my above comments, I will still intend to keep my original assessment. I thank the authors for the in-depth thoughts exchange.

---

> > > ### Author Response · Authors · 2021-11-30
> > > **Further Response to Clarify Misunderstandings**
> > >
> > > Dear Reviewer 5DSq,
> > >
> > > We appreciate your detailed comments. We want to clarify some misunderstandings that caused some of your concerns.
> > >
> > > * **Comparison to MLP + C&S**
> > >
> > > We agree C&S is a strong model, and “MLP+C&S can already achieve 100 times faster for training and inference time, and has 137 times fewer parameters”. However, we want to clarify that these advantages are under the **transductive setting**. C&S by its nature doesn’t work for inductive predictions (unless one runs LP for the entire graph each time for predicting a new node, which is not reasonable). Also, the “100 times faster” is regarding the **total time for prediction on all nodes in the graph**. If we only want to predict a few nodes, C&S can be even slower than GNN, where the former runs LP over the entire graph, and the latter only does local L-hop message passing. In our case, we show both the transductive and inductive settings, but our major target use case is the **inductive setting**. We don’t think prediction over the entire graph is done very frequently, but the inductive prediction can happen frequently for many applications and lead to throughput issues mentioned in **point 4 in our first response**, which is why the inference acceleration is needed. For these reasons, although both “solve a similar time computation efficiency issue”, C&S is related but not super similar and directly comparable to GLNN.
> > >
> > > * **“The author claims that using a stronger GNN teacher model’s prediction label to guide the learning of a weak student model shows improvement over the vanilla MLP student model is a surprising discovery"**
> > >
> > > To clarify, we don’t intend to claim knowledge distillation (KD) itself as our contribution. Rather, a core tenet of our contribution is that we show that KD from a relational model (GNN) to a non-relational model (MLP) works shockingly well and has powerful practical implications. It suggests that the use of relations/edges in inference may not be nearly as necessary as we (or the larger community) thought. We don’t necessarily have to tolerate slow neighbor fetching during inference to enjoy the performance benefits of relational models. We don’t feel it is obvious that student MLPs can in general be competitive to teacher GNNs, as none of the prior KD works [3-7] consider the relational-to-non-relational context.
> > >
> > > * **"A justified claim regarding alternative scalable GNN designs, … node-wise sampling is a de-facto is not true"**
> > >
> > > In **point 3 of our first response**, we have already shown that Cluster-GCN and GraphSAINT don’t accelerate inference. Other related sampling techniques we are aware of, like LADIES, FastGCN, and etc, also don’t accelerate inference or can only do as well as node-wise sampling. Therefore, we mentioned node-wise sampling as the de-facto approach in practice, as to the best of our knowledge, it is a predominant theme in literature (Pinterest [8], Uber [9], Ant Financial [10], and Snapchat [11]). Certainly, we agree that this does not mean that _every_ industrial deployment of GNNs/associated relational approaches will use these exact same settings. We would appreciate other advanced sampling techniques used in practice if you can point us to them.
> > >
> > > * **Caching the sampled neighbor for each node can reduce the inference time**
> > >
> > > First of all, for the inductive setting that we majorly focus on, the test node is new, and it is not possible to cache its neighbor nodes in advance.
> > >
> > > For the transductive setting, we agree that caching might be a part of an optimal practical solution. However, while caching can speed up and lighten the practical cost of fetches (e.g. lookups from 100ms to 10ms), they do not change the ultimate time complexity. Moreover, they bring extra space concerns, which require localized low-latency caching solutions (e.g. main memory) during inference on top of the main cloud-based relational stores for large graphs. This is a big disadvantage and is infeasible for extremely large graphs in practical settings.
> > >
> > > * **“GNN on a resource constraint device/case are limited. The urging requirement for speedup GNN is on the training phase, not the inference phase, such as recommendation systems (RS)”**
> > >
> > > For such scenarios, we believe we have already given a detailed discussion with examples in **point 4 of our first response**, including the benefits of the inference speedup for RSs [14], and the loss of revenue and worsened user experience with increased latency [12, 13].
> > >
> > > While we don’t disagree that speedup GNN training is valuable, we feel it is not quite accurate to suggest that speedup GNN inference is insignificant or unimportant. Considerable focus is invested in making accurate ML models fast to deploy for inference [15] — we don’t feel GNNs are an exception; rather, they have _additional_ relational dependence that must be carefully managed, lest it blows up inference time.
> > >
> > > [15] Minimizing real-time prediction serving latency in machine learning, Google Cloud

---

> ### Author Response · Authors · 2021-11-17
> **Response (2/3)**
>
> 3. Batching and sampling techniques reduce node fetching, e.g. Cluster-GCN [8] and GraphSAINT [9]
>
> * Firstly, we clarify that $O(R^L)$ is for general full-neighbor fetching at the **inference** stage. Batching and sampling techniques, e.g. [8,9], were proposed for faster GNN **training**. Like SAGE, they can be combined with GLNN for GNN teacher training. They are indeed faster than $O(R^L)$ in training, although there is still an exponential dependency in the node degree and number of layers. For inference, however, they resort to the full-neighbor fetching due to their nature, which we discuss next:
>
>
> * According to Sec 6.2 in Cluster-GCN, to inductively infer a new node, it needs to fetch nodes from the **original graph** instead of the **partitioned graph** because the graph partitioning was only applied for the training graph. It won’t be worth re-do the partitioning for each new node. Without partitioning, Cluster-GCN is essentially a full-neighbor GCN.
>
>
> * The main point of GraphSAINT is to sample **many** subgraphs for each target node $v$ and properly normalize the contribution of all nodes $u$ that pass messages to $v$ during **training**. As shown in Algorithm 1 and Sec 3.2 in GraphSAINT, both training and the normalization factor computation need many subgraphs (200 in the official code). This step only needs to be done once for training, but for inductive inference, full-neighbor fetching is preferred to avoid sampling many subgraphs for each new node.
>
>
> * Lastly, we briefly note that node-wise sampling is still a common/de-facto approach in many practical scenarios, e.g. for Pinterest [8], Uber [9], Ant Financial [10], and Snapchat [11].
>
> 4. Scenarios we can’t meet the inference time constraint even with batching and graph-wise sampling
>
> * As discussed in response #3, batching and graph-wise sampling are typically for training. Inference efficiency remains a challenge even with node/layer-wise sampling. More importantly, instead of meeting a fixed constraint, practical (deployed) models almost always aim to minimize inference time. For GNNs, due to the graph-dependency nature, inference time can meet a fixed time constraint for now doesn’t mean it can meet the constraint later when the user base grows and graph densification occurs. A concrete example is GraphInfer [10]: Table 5 in GraphInfer shows that it takes 4423s for a full batch inference of the highly-optimized GraphInfer, so for each 0.01% nodes growth, it will take an added ~400ms, which is considered slow in the standard of the Amazon study: “every 100ms of latency cost them 1% in sales” [12].
>
>
> * Moreover, from the computing side, inference time can meet a constraint on a server doesn’t mean the same model can meet the constraint when deployed on a phone. From the throughput side, fast inference is necessary for high-throughput. When #inference query increases, a small improvement on each query can greatly affect business revenue. A related study by Akamai shows that "A 100-millisecond delay in website load time can hurt conversion rates by 7 percent" [13]. Another example where GNNs are used more often is recommendation systems (RSs). Real RSs often go through two steps. Stage A retrieves representations of relevant items. Then stage B ranks over them [14]. The lower latency in stage A, the more items can be retrieved, and the better performance in stage B for the overall ranking.
>
> 5. GLNN is not competitive on Arxiv and Products.
>
> * The referred results for Arxiv (63.5%) and Products (68.9%) are from Table 1 under the setting of controlling space complexity where the teacher and the student have the same number of parameters. We include this setting as many distillation works focus on space complexity. In our case, we care more about time complexity than space complexity. Therefore, in Table 2, we increase the student size to get GLNN+, which can achieve 72.15% on arXiv and 77.65% on Products and are competitive to the SAGE teacher (70.92% and 78.61%).
>
>
> * Space complexity in terms of #parameters is a common consideration for distillation, but it is actually not fair for comparing GNNs and MLPs. For GNN inference, the graph needs to be loaded in memory entirely or in batches, which often takes a much larger space than model parameters. Thus, the space complexity of GNNs is higher than equal-parameter MLPs owing to the graph “burden”. In our case, the largest GLNN+ only has 3 layers and 2048 hidden dim, which is small compared to 2.5M nodes in Products.
>
>
> * From the time complexity side, inference latency of GNNs majorly comes from the data dependency (Sec 4). We show in Figure 3 that a 5-layer MLP with 8X hidden dim runs much faster than a 1-layer SAGE. From Table 1 to Table 2, GLNN+ on Products maintains accuracy (0.96 drop) but achieves a large time gain (273X faster) as in the table below, which shows the trade-off is beneficial in practice.
> ||SAGE|GLNN+|
> |-|-|-|
> |Acc|78.61|77.65 (-0.96)|
> |Time(ms)| 2071.3|7.56 (273X)|

---

> ### Author Response · Authors · 2021-11-17
> **Response (1/3)**
>
> Dear Reviewer 5DSq,
>
> We really appreciate your comments. We hope our point-to-point response can address your concerns and clarify our contribution.
>
> 1. Limited novelty, distillation formulation is from Hinton et al. 2015.
>
> * As an exploratory work, our goal is to reveal that combining existing techniques can lead to surprisingly good results and strongly benefits real-world use cases. There are extensive examples of valuable exploratory papers that leverage existing ideas to great practical impact: for example, the well-recognized and exploratory distillation work from Hinton et al. 2015 [1] also leverages the same technique of learning from teacher-generated pseudo-labels from Bucila et al. 2006 [2].
>
>
> * We believe the value of exploratory papers, including ours, comes from identifying surprising results of the method (Sec 5.3, 5.4), quantifying its advantages over other methods (Sec 5.5), analyzing the method properly (Sec 5.6, 5.7, 6), pointing out its limitations (Sec 5.8). We feel we have properly shown these points with various experiment supports. For example, in Sec 5.6, we study what GLNN actually learned from teacher GNNs. Through the min-cut loss measure, we found that GLNN predictions after distillation are much more consistent with graph topology than predictions of a standalone MLP. This consistency is an inductive bias specific to GNNs but not MLPs. Our analysis shows that one plausible reason for GLNNs to perform well is that the inductive bias was transferred to GLNN via distillation. Our work is the first to show that such an inductive bias transfer from a GNN to an MLP can massively improve the MLPs generalizability for a key graph ML task like node classification, while preserving its scalability which GNNs cannot enjoy (by design) at inference time.
>
>
> * In contrast, our work does not aim to exactly be a “new-method paper,” even though our methodological proposal is indeed a novel choice in literature with significant practical implications. We do not propose further complexities in the model design, as it would conflict with the message we try to convey - an uncomplicated design like GLNN can achieve great results for most node classification benchmarks with practical benefits.
>
>
> * For these reasons, we feel it is a bit unfair to claim that “The contributions are neither significant nor novel” in both technical and empirical significance.
>
>
> 2. Missing discussion of related work [3], [4], [5]
>
> * Thank you for pointing out these related works. In our initial version, some papers specific to GNN distillation are cited in the related work section but not discussed in detail due to the space limit, e.g. [3], [6], [7]. We deferred the discussion of these works to the second paragraph of Sec 5.5 when we compare different inference acceleration methods in detail. Nevertheless, we have added an additional paragraph in the related work section for GNN distillation including [3-7]. We show the new paragraph below and highlight it in red in our updated pdf.
>
>
> * “GNN distillation. Several previous works tried to distill large teacher GNNs to smaller student GNNs. We now introduce these works and clarify the difference between them and ours. LSP [3] and TinyGNN [6] are similar, in the sense that they both distill a large GNN to some smaller student GNN while preserving local neighbor information. The student model in LSP and TinyGNN is a GNN with fewer parameters, but not necessarily fewer layers than the teacher model, which makes both designs still depend on the graph data and require latency-inducing fetches. Distillation in GFKD is done via generating fake graphs [4]. GFKD focused on graph-level prediction tasks, for which data instances are independent graphs. In our case, we consider node-level tasks and focus on removing dependency between data instances at inference time. GraphSAIL [5] uses distillation for GNN incremental learning. Instead of efficiency, the goal of GraphSAIL is to train a student that works well on new data while preserving model performance on the old data via distillation. More recently, CPF [7] proposed to distill GNNs to a student model that includes label propagation to improve the GNN performance. CPF is no longer distilling to a GNN, but it is still heavily graph-dependent as it uses label propagation. In contrast, we are removing the graph dependence and distill to simple MLP students.”

---

> ### Author Response · Authors · 2021-11-29
> **Need More Clarification?**
>
> Thank you for your feedback. We feel like we have addressed all of your questions from the initial reviews in our response. As we are approaching the end of the discussion period, we would like to ask whether there are any remaining concerns regarding our paper or our response? We are happy to answer any further questions.

---

### Decision · Program_Chairs · 2022-01-20

**Decision:**

Accept (Poster)

**Comment:**

This paper proposes a very simple procedure to accelerate the inference time of graph-structured Neural Networks, by distilling knowledge of a GNN into a node-wise MLP.
Despite some concerns about the novelty of the methodology (which borrows heavily from previous KD works), reviewers generally found this empirical work well executed and providing a potentially useful baseline for large-scale applications. Therefore, the AC recommends acceptance.